# Wnt/β-catenin signalling modulates the timing of cell fate decision making in the early mouse embryo

Joaquin Lilao-Garzón[1,2¤a], Elena Corujo-Simon[2¤b], Meritxell Vinyoles[3¤c], Sabine C. Fischer[4], José Guillén[1], Tina Balayo[1¤d], Silvia Muñoz-Descalzo[1,2]*

1 Instituto Universitario de Investigaciones Biomédicas y Sanitarias, Universidad de las Palmas de Gran Canaria, Las Palmas de Gran Canaria, Spain, 2 Department of Biology & Biochemistry, University of Bath, Bath, United Kingdom, 3 Department of Genetics, University of Cambridge, Cambridge, United Kingdom, 4 Julius-Maximilians-Universität Würzburg, Faculty of Biology, Center for Computational and Theoretical Biology, Würzburg, Germany

¤a Janelia Research Campus, Howard Hughes Medical Institute, Ashburn, United States of America
¤b MRC Human Genetics Unit, Institute of Genetics and Cancer, University of Edinburgh, Western General Hospital, Edinburgh, United Kingdom
¤c Josep Carreras Leukemia Research Institute, Department of Biomedicine, School of Medicine, University of Barcelona, Barcelona, Spain
¤d Systems Bioengineering, MELIS, Universidad Pompeu Fabra, Barcelona, Spain
* silvia.munoz@ulpgc.es

## Abstract

Cell fate choice is a key event happening during preimplantation mouse development. From embryonic day 3.5 (E3.5) to E4.5, the inner cell mass (ICM) differentiates into epiblast (Epi, NANOG expressing cells) and primitive endoderm (PrE, GATA6, SOX17 and/or GATA4 expressing cells). The mechanism by which ICM cells differentiate into Epi cells and PrE cells remains partially unknown. FGF/ERK has been proposed as the main signalling pathway for this event, but it does not explain co-expression of NANOG and GATA6 or how the cell fate choice is initiated. In this study, we investigate whether Wnt/β-catenin signalling also plays a role. To this end, we use two *in vitro* models based on inducible GATA6 expression: one in 2D (flat cultured cells), and another in 3D, namely ICM organoids. By combining these *in vitro* models with *in vivo* mouse embryos, chemical and classical genetics, and quantitative 3D immunofluorescence analyses, we propose a dual role for Wnt/β-catenin signalling. We find that β-catenin, acting alongside FGF/ERK signalling, helps to guide the cell fate choice towards PrE. Additionally, by regulating GATA6 and GATA4 stability, Wnt/β-catenin signalling further facilitates this choice. To summarise, we observe that Wnt/β-catenin signalling pathway activation promotes PrE differentiation, while its inhibition delays it.

**Data availability statement:** https://zenodo.org/records/17199526.

**Funding:** Early work at the University of Bath was supported by a Wellcome Trust Seed Award (109589/Z/15/Z) and ECS funding was provided by the University of Bath. At the ULPGC, work at SMDlab was funded by the ACIISI (CEI2019-02), Programa de Ayudas a la Investigación de la ULPGC, and ACIISI co-funded by FEDER Funds (ProID2020010013). JLG was supported by the ULPGC predoctoral program and SMD was supported by the "Viera y Clavijo" Program from the ACIISI, and the ULPGC. Imaging at the IUIBS was done at the SIMACE (Servicio de Investigación en Microscopía Avanzada Confocal y Electrónica). Mouse work at the IUIBS was done at the now established SABiA (Servicio de Animalario y Bienestar Animal). TB was funded by a "Ayuda para Personal Técnico de Apoyo" from the Ministerio de Economía Industria y Competitividad (PTA2017-14230-I). Work at the SCF lab was supported through funding by the Deutsche Forschungsgemeinschaft (DFG, German Research Foundation) project number 470129398 and start-up funding by the University of Wuerzburg. MV funding was provided by EMBO Postdoctoral fellowships (ALTF 509-2015).

**Competing interests:** The authors have declared that no competing interests exist.

## Introduction

Early preimplantation development of the mammalian embryo involves two sequential cell fate decisions that equip the embryo with the essential machinery for implantation and, subsequently, successful development until birth (reviewed in [1]). During the first decision, a subset of cells differentiate into trophectoderm (TE) that will later form the embryonic portion of the placenta. The other cells (so-called inner cells mass, ICM, cells) will differentiate into epiblast (Epi, will become the embryo proper) or primitive endoderm (PrE, will form the yolk sac). Multiple studies have addressed how ICM cells differentiate into Epi or PrE cells. In mice, during the differentiation process, NANOG and GATA6 are co-expressed in ICM cells [2–9]. Their mutual inhibition results in Epi or PrE fate differentiation: Epi cells maintain NANOG expression (downregulating GATA6 expression), while PrE cells keep GATA6 expression (downregulating NANOG expression) [3,10]. At later stages, PrE cells express other fate markers like GATA4, SOX17 and SOX7 [11], and sort facing the embryo cavity. Our recent research has utilised single-cell quantitative immunofluorescence analysis (QIF) alongside three-dimensional neighbourhood analyses and mathematical modelling. This approach has underscored the importance of investigating cell fate decisions within the context of the entire ICM [12,13]. Furthermore, we demonstrated that maternal factors such as age, obesity, and hyperglycaemia are associated with delays in these cell fate decisions [14].

The FGF/MAPK signalling pathway is the primary mechanism driving cell fate decision, as it promotes PrE fate while inhibiting Epi fate [8,10,15–21] (reviewed in [22]). However, other signalling pathways also play significant roles. Specifically, active p38-Mapk14/11 is required for PrE differentiation [23]. Additionally, PI3K/AKT signalling is active during preimplantation development, regulates NANOG and enables cells to respond to FGF [24,25]. We have previously hypothesised a potential role for Wnt/β-catenin signalling in this process [26] based on its function during the pluripotency and differentiation of the *in vitro* ICM counterpart—mouse embryonic stem cells (mESCs) [26–28]. In mESCs, β-catenin stabilises the pluripotent state by forming a complex with NANOG and OCT4 localised at the cell membrane; during differentiation, the complex is disassembled, and β-catenin is free to enter the nucleus to promote the transcription of differentiation-related genes [27–29].

Previous studies have examined the potential role of Wnt/β-catenin *in vivo* during mouse preimplantation development. Maternally deposited β-catenin in the mouse oocyte is sufficient to successfully complete preimplantation development [30]. Consequently, materno-zygotic *β-catenin* mutant embryos were generated to study its function [31]. Initial studies suggested that the traditional mutant allele produced a truncated version of β-catenin [31–33]. For this reason, a new null allele was generated, and materno-zygotic *β-catenin* mutant embryos were analysed [33]. These embryos showed defects in blastomere adhesion and size but did not display qualitative defects in lineage allocation. Likewise, other studies addressing a potential role for Wnt/β-catenin signalling during mouse preimplantation development using qualitative methods found no evidence of involvement in either blastocyst formation or cell fate allocation [34,35]. Regardless, several studies in mouse embryos and mESCs

indicated that activation of the pathway promoted PrE fate [36–38]. Another study points towards activation of the pathway leading to Epi fate [39]. Others suggested a role for Wnt/β-catenin signalling in the ICM based on reporter experiments [40] that could not be replicated in alternative more sensitive single cell reporters [41]. Hence, the involvement of Wnt/β-catenin signalling during mouse preimplantation development remains controversial.

Here, we explore the role of Wnt/β-catenin signalling during Epi vs PrE differentiation using quantitative methods. We use a combination of *in vitro* (2D and 3D) and *in vivo* models, alongside chemical and genetic modulation, coupled with data analyses based on quantitative immunofluorescence. Our detailed single cell quantitative analyses in multiple models allows us to propose that Wnt/β-catenin signalling promotes PrE fate, playing a role during cell fate allocation in mouse preimplantation embryos.

## Results

### Membrane β-catenin is higher in Epi precursors compared to PrE precursors

During mouse embryonic stem cells (mESCs) differentiation membrane localised β-catenin is released and transcriptional activity is detected [28]. Given the embryonic origin of mESCs, we investigated the subcellular localisation of β-catenin in mouse preimplantation embryos, focusing on its relationship with the epiblast (Epi) and primitive endoderm (PrE) fate markers NANOG and GATA6, respectively (Figs 1, 2). In early (E3.5, 32–64 cells) and mid (E4.0, 65–90 cells) blastocysts, we observe high membrane β-catenin levels in ICM cells (cells co-expressing both NANOG and GATA6, N + G6 + , Fig 1A-B). In late blastocysts (E4.5, > 90 cells), β-catenin levels remain high in the Epi cell (N + G6-) membranes and is downregulated in PrE (N-G6+) cell membranes (Fig 1C). No obvious nuclear β-catenin can be observed, not even using a specific antibody against the transcriptionally active form of β-catenin (S1 Fig).

To examine possible differences depending on the developmental stage and cell fate decision progression, we measured membrane β-catenin levels between ICM (N+G6+), Epi (N+G6-) and PrE (N-G6+) cells in early, mid and late embryos (Fig 2). In early embryos, all N + G6 + cells exhibit high membrane β-catenin levels (Fig 2A-C). In mid blastocysts, membrane β-catenin levels are higher among adjacent N + G6- than between adjacent N + G6- cells and N-G6+ or adjacent N-G6 + cells (Fig 2D-F). By late embryos, the differences observed in mid-blastocysts become more pronounced (Fig 2G-I).

Altogether, these results align with our previously published β-catenin subcellular location in mESCs [28]: undifferentiated cells (ICM cells) have high levels of membrane β-catenin with no clear nuclear localization, while for differentiated cells, Epi cells retain high membrane β-catenin levels, whereas PrE cells show a decrease in membrane β-catenin.

### Chemical modulation of Wnt/β-catenin signalling influences PrE/Epi fate *in vitro*

Our previous results in mESCs show that Wnt/β-catenin signalling is activated during differentiation [27,28]. Additional studies in mouse embryos and mESCs also indicated that activation of the pathway promoted PrE fate [36–38]. Altogether, this led us to propose that it may play a role in PrE differentiation [26].

To test the involvement of Wnt/β-catenin signalling in PrE differentiation, we used *tet::Gata6*-mCherry mESCs [38]. Briefly, to induce differentiation, NANOG expressing *tet::Gata6*-mCherry mESCs are treated with doxycycline (dox) for 6 hours to induce the expression of *Gata6*-mCherry allowing the cells to acquire an ICM-like state co-expressing NANOG and GATA6 (Fig 3A). Following this induction, cells are cultured in mESCs medium without dox for 24 hours to allow their differentiation into Epi-like or PrE-like cells. To test the involvement of Wnt/β-catenin signalling in PrE- versus Epi-like fate differentiation, we cultured these cells throughout the experiment in the presence of Chi (a well-known Wnt/β-catenin signalling activator), or XAV (an inhibitor) (Fig 3 and S2A [42,43]). To confirm induction of the PrE programme, we monitor SOX17 (S17) and GATA4 (G4) expression with quantitative immunofluorescence (QIF) analysis followed by population analysis [12,44]. We observe a clear effect in PrE differentiation after modulating Wnt/β-catenin signalling. Pathway activation increases the percentage of PrE-like cells (N-S17+ or N-G4+), which also exhibited higher expression levels of the PrE markers SOX17 and GATA4

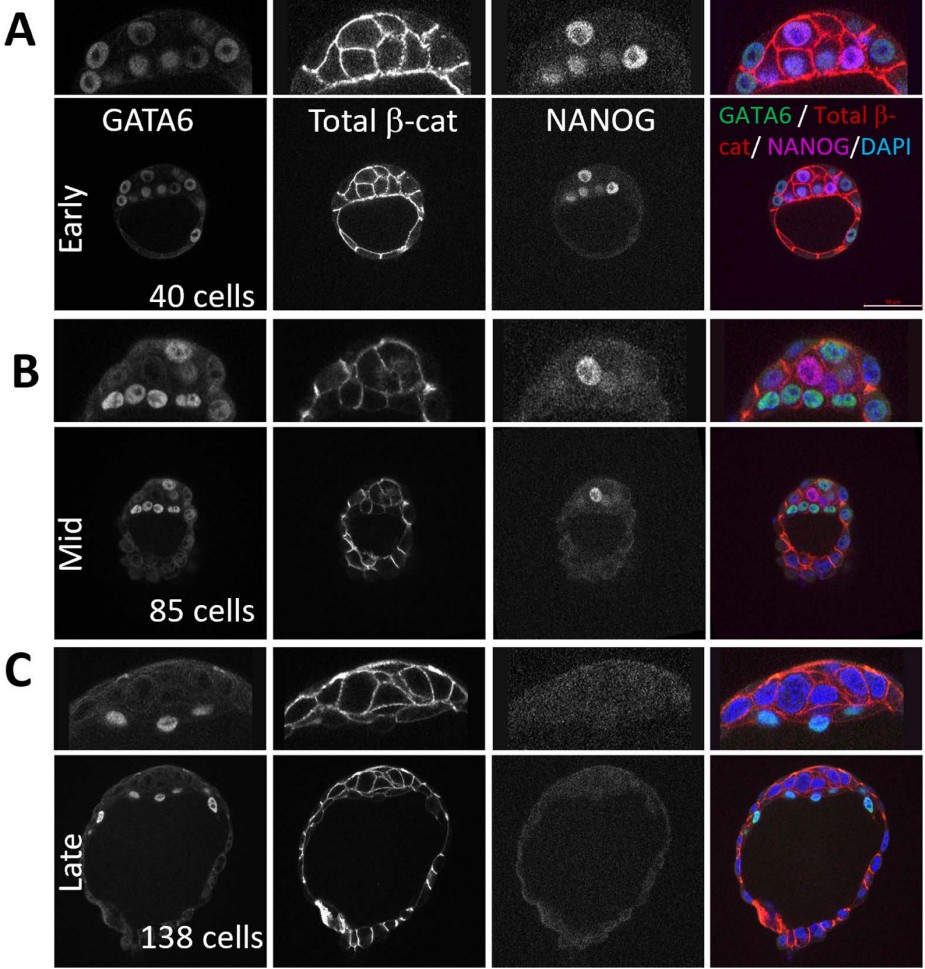

**Fig 1. β-catenin localises in cell membranes in mouse preimplantation embryos.** Representative single confocal images of early, mid and late mouse blastocysts stained with antibodies against GATA6 (green), total β-catenin (red) and NANOG (magenta), DAPI (blue) was used to stain nuclei. Higher magnification images are above the whole embryo images. Scale bar: 50 μm.

(Fig 3B-C and S2B). Conversely, pathway inhibition produced opposite effects: a reduced percentage of PrE-like cells. The effects on the other cell-types are less pronounced: ICM-like are influenced by signalling activation and inhibition when monitoring N+S17+cells, and only by signalling activation, but not by inhibition, in N+G4+cells. The percentage of N-S17-cells increase under both conditions, and N-G4- cells increase only under signalling activation. These cells might be in an advanced Epi-like fate where NANOG has been already downregulated, as previously suggested [9]. However, since these cells did not express any of the assessed markers, their fate remains ambiguous.

These results suggest that activation of Wnt/β-catenin signalling enhances PrE differentiation efficiency, whereas its inhibition hinders this process.

### Wnt/β-catenin chemical modulation influences PrE/Epi fate *in vivo*

To examine whether modulation of Wnt/β-catenin signalling impacts PrE differentiation *in vivo*, we cultured early mouse preimplantation embryos (E3.5) in the presence of Wnt or XAV for 24h (Fig 4 and S3) and monitored PrE differentiation

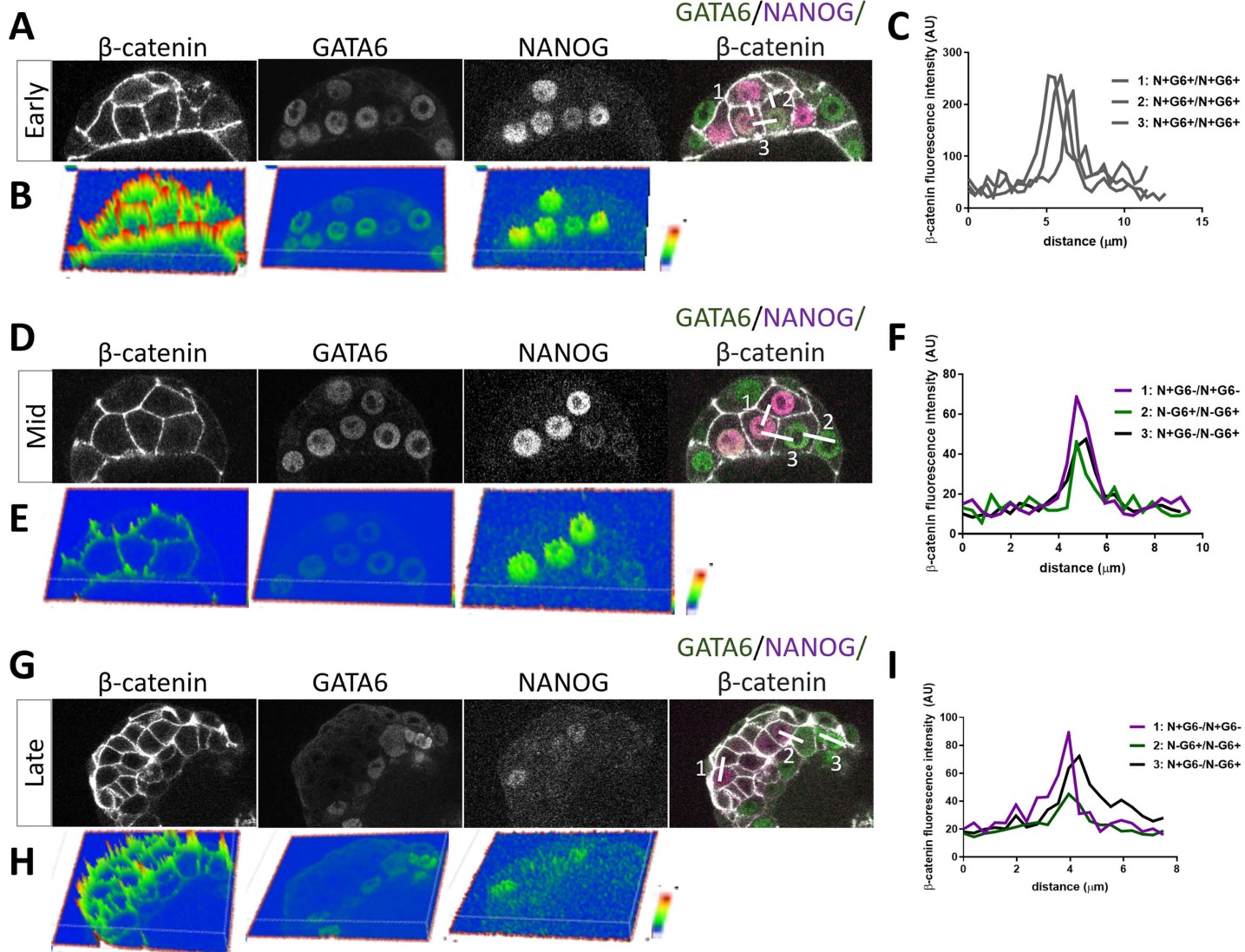

**Fig 2. Membrane β-catenin quantification during cell fate acquisition. (A, D** and G) Representative single confocal images of early, mid and late mouse blastocysts stained with antibodies against total β-catenin (white), GATA6 (green), and NANOG (magenta), DAPI (blue) was used to stain nuclei. Scale bar: 50 μm. **(B, E** and H) Fluorescence intensity representation from the confocal images using ARIVIS software. The heat map represents the intensity of the indicated markers (blue for low, green for medium, red for high). **(C, F** and I) Profile plots showing the variation in fluorescence intensity across the numbered lines drawn shown in the merged panel. Grey lines show the profile between ICM (N+G6+) cells, magenta lines are between Epi cells (N+G6-) cells, green lines are between PrE cells (N-G6+) cells, and black lines are between Epi and PrE cells (N+G6-/N-G6+).

using GATA4 as a marker. Consistent with *in vitro* observations, activation of Wnt/β-catenin signalling results in a higher proportion of PrE cells accompanied by increased GATA4 levels, at the expense of Epi cells (Fig 4A-B and S3, upper panels). Conversely, inhibition of signalling with XAV reduced PrE differentiation, leading to an increase in Epi cells with higher NANOG levels (Fig 4C-D and S3, lower panels). ICM cell proportion is also increased.

These results demonstrate that, *in vivo*, PrE and Epi fate decisions are also regulated by Wnt/β-catenin signalling. Activation of the pathway promotes PrE differentiation while hindering Epi fate. Conversely, inhibition decreases PrE differentiation and favours Epi fate.

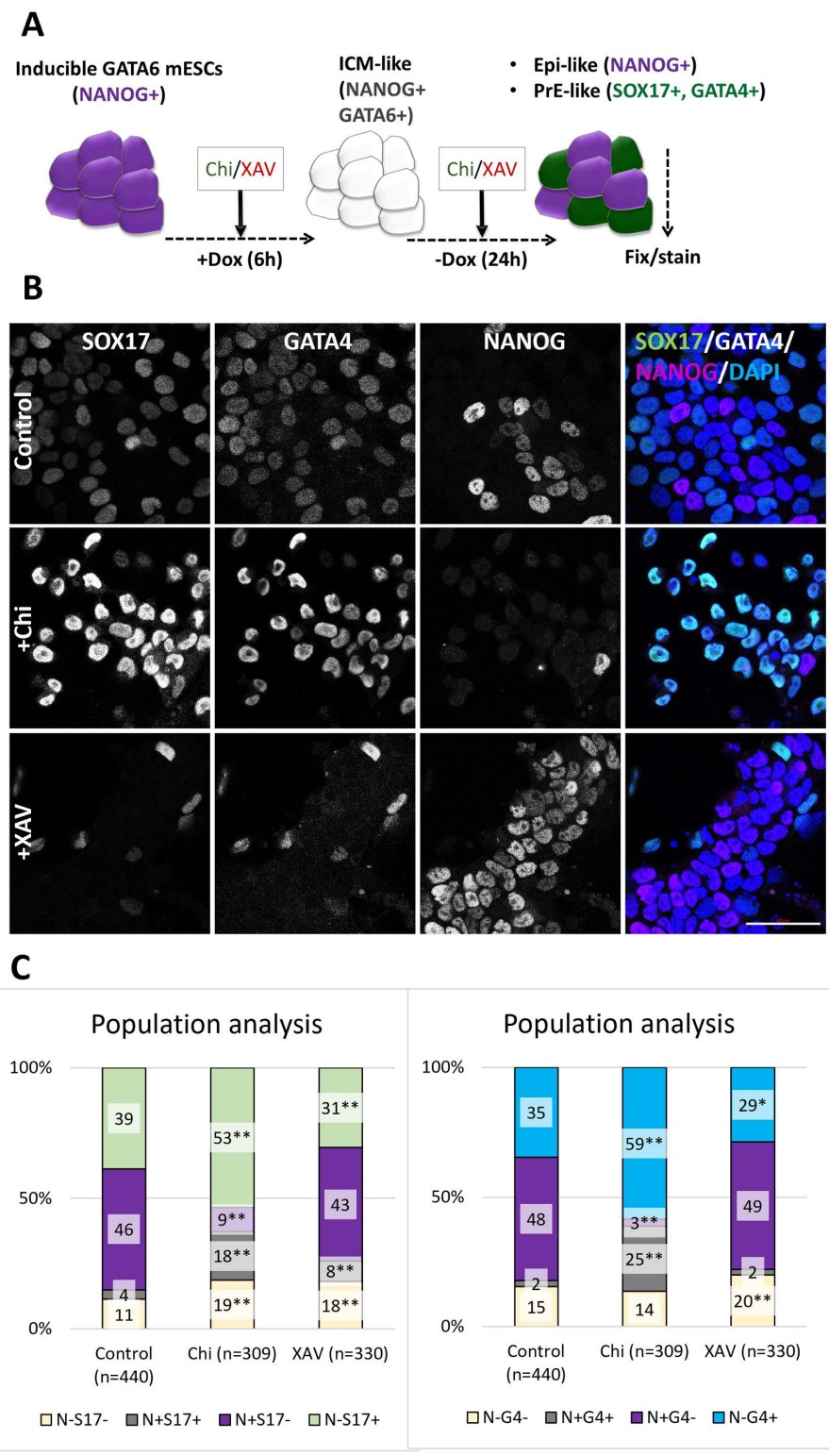

**Fig 3. Chemical Wnt/β-catenin signalling modulation influences PrE fate in vitro.** (A) Treatment regime of *tet::Gata6*-mCherry mESCs. (B) Representative single confocal images of control (first row), Chi treated (central), or XAV treated (bottom) cells stained with antibodies against SOX17 (green), GATA4 (white) and NANOG (magenta), DAPI (blue) was used to stain nuclei. Scale bar: 50 µm. (C) Population analyses show the mean percentage of

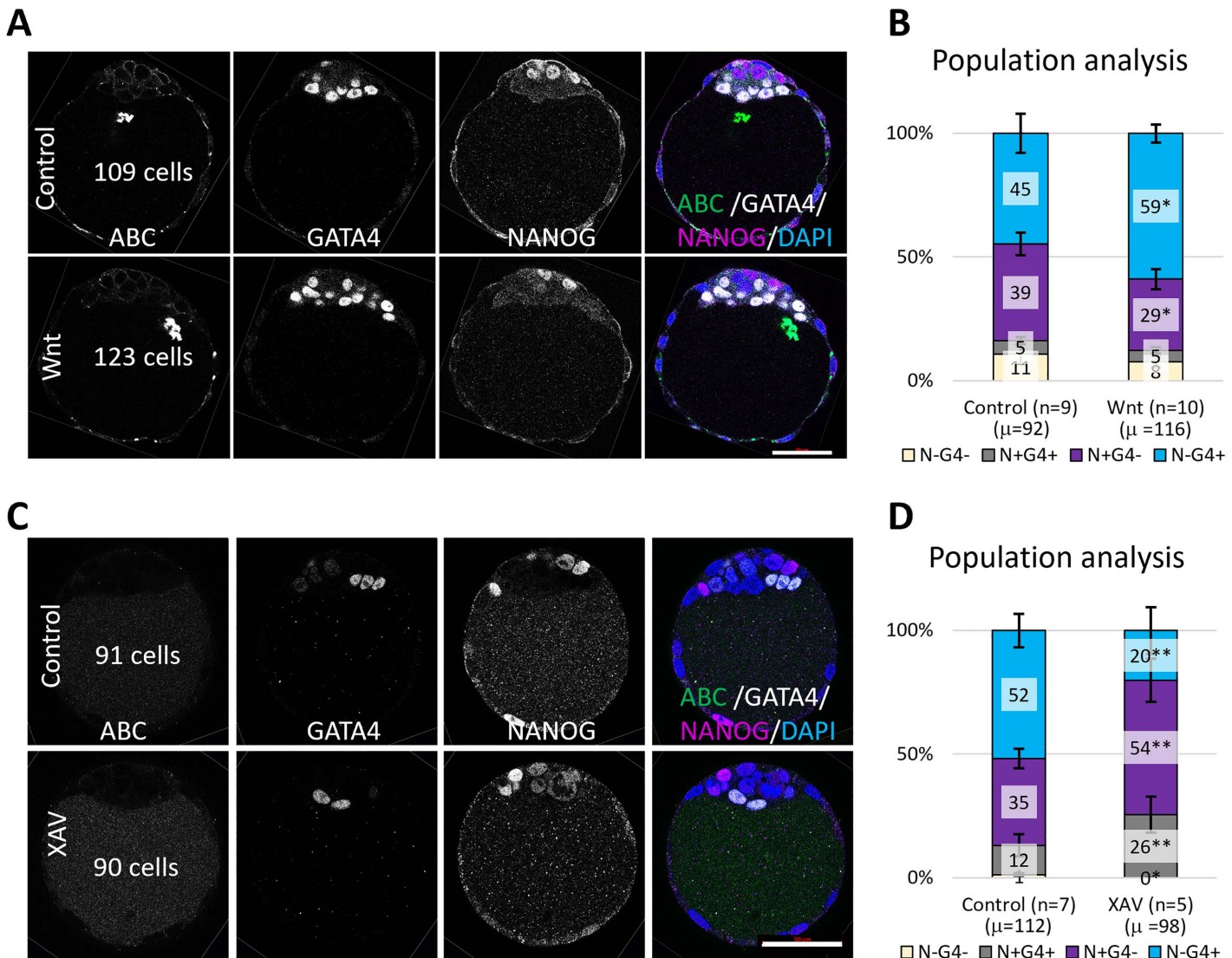

**Fig 4. Chemical Wnt/β-catenin signalling modulation influences PrE fate in vivo.** (A, C) Representative single confocal images of control (first rows), and Wnt3a- or XAV-treated (bottom rows) early mouse embryos (E3.5) for 24h and stained with antibodies against active β-catenin (ABC, green), GATA4 (white) and NANOG (magenta), DAPI (blue) was used to stain nuclei. Scale bar: 50 μm. (B, D) Population analyses show the mean percentage of N-G4-, N+G4+, N+G4- and N-G4+cells. Error bars indicate the s.e.m. One-tailed unpaired t-test was used to compare cell populations´ distribution between control and treated embryos. *:p<0.01; **: p<0.05. n is the number of analysed embryos. μ is the average embryo total cell number.

## Wnt/β-catenin signalling genetic inhibition influences PrE/Epi fate *in vitro*

To further investigate the effect of Wnt/β-catenin signalling on PrE differentiation, we generated new *β-catenin* mutant lines in the *tet::Gata6*-mCherry mESCs background [38] (Fig 5, and S4-S5). We used commercially available CRISPR gRNAs

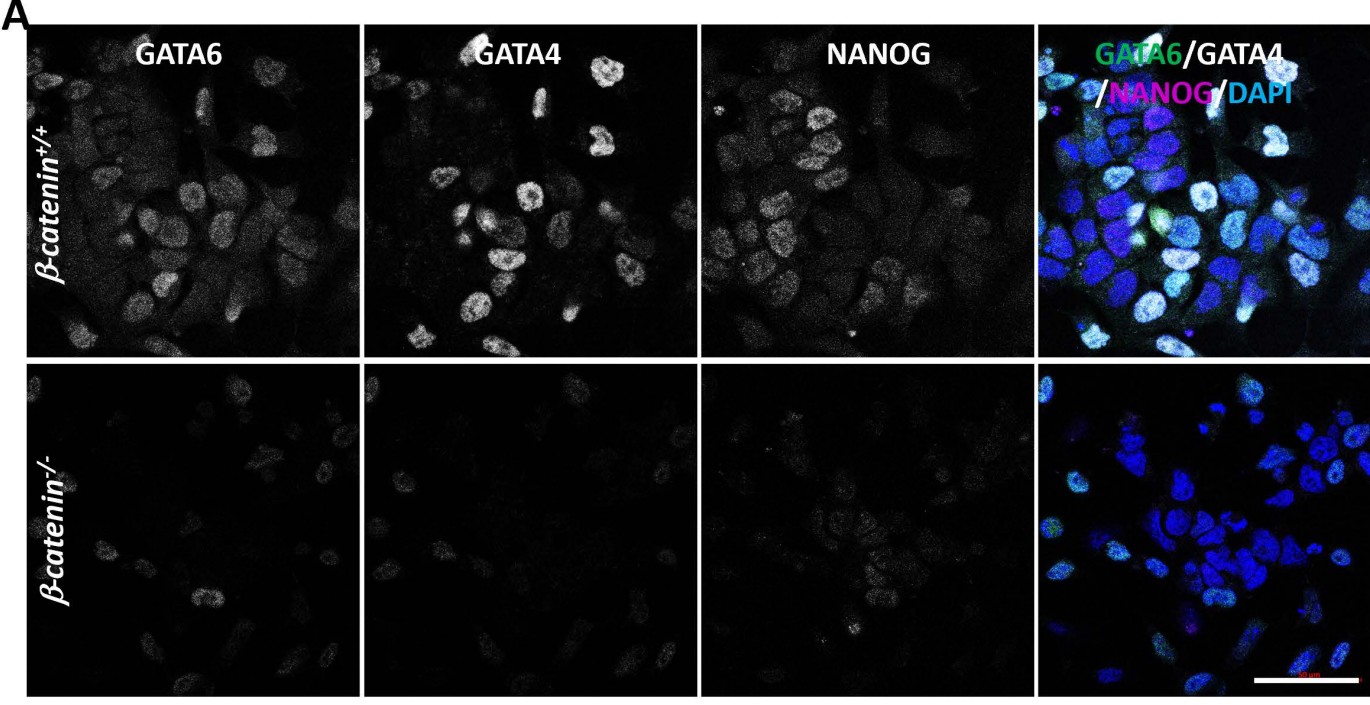

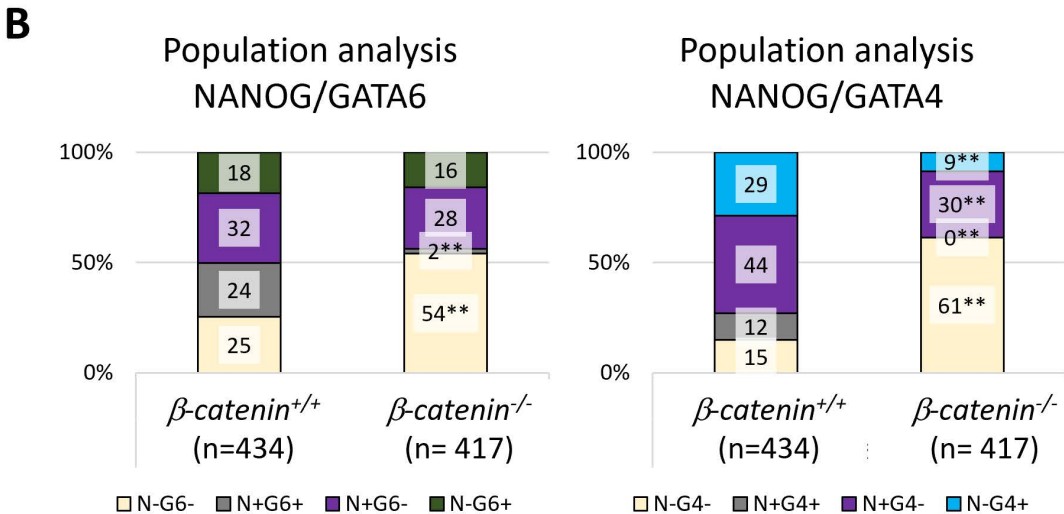

**Fig 5. Genetic Wnt/β-catenin signalling inhibition hinders PrE fate in vitro (2D).** (A) Representative single confocal images of β-catenin⁺/⁺ *tet::Gata6*-mCherry (above) and β-catenin⁻/⁻ *tet::Gata6*-mCherry (below) mESCs after inducing PrE differentiation and stained with antibodies against GATA6 (green), GATA4 (white) and NANOG (magenta), DAPI (*blue*) was used to stain nuclei. Scale bar: 50 μm. (B) Population analyses show the mean percentage of N-G6-, N+G6+, N+G6- and N-G6+ (left) or N-G4-, N+G4+, N+G4- and N-G4+ (right) cells. Two tailed-Z-test, **: $p < 0.05$.

plasmids (see M&M). Several clones were generated and two (C5 and F1) were selected for further analyses (S4A-E Fig). C5 was sent for sequencing, and it has a deletion spanning exons 4–5 (S4F Fig), resulting in no detectable functional protein (S4G-H Fig). As with the chemical inhibition of the pathway, we observe a decreased efficiency in PrE-like

differentiation, but only when using GATA4 as a marker. Specifically, there is a lower proportion of PrE-like cells (N-G4+, Fig 5B) and decreased expression levels of both GATA6 and GATA4 (S5 Fig). In the absence of β-catenin, we also detect a reduction in the proportion of Epi-like cells (N+G4-) and NANOG expression levels, likely due to its effect on NANOG stability or indicating an advanced Epi state [9,27]. There is also an increase in N-G6- (or N-G4-) cells that, as previously suggested, may reflect more advanced Epi-like cells. This would also be reflected in the increase in NANOG only positive cells when using the more advanced PrE marker (GATA4), but not with the earlier marker (GATA6).

We next used the newly generated *β-catenin* mutant line to generate ICM organoids, a three-dimensional *in vitro* system that mimics ICM differentiation into Epi and PrE [45] (Fig 6 and S6-S7). *β-catenin* mutant ICM organoids exhibit no statistically significant differences in PrE/Epi-like differentiation (Fig 6A-C, regimes I, control, and II, mutant). However, we observe a defect in NANOG expressing cells upon analysing the 3D distribution of Epi- and PrE-like cells. Unlike wild-type ICM organoids, where high NANOG-expressing cells are found closer to the organoid centroid (Fig 6D, black line [45]), some high NANOG-expressing cells are in the periphery (Fig 6D, red line). The complete absence of *β-catenin* does not affect the total cell number in the ICM organoids (S7A Fig). Here we do not observe differences in NANOG or GATA6 expression levels (S7B Fig) nor in GATA6 expressing cells distribution (S7C Fig).

We next challenged the differentiation by generating ICM organoids chimeras by mixing *β-catenin* wild-type and mutant cells. To achieve this, we induce GATA6 expression in both cell lines prior to mixing (Fig 6A, regime III), while using β-catenin antibody staining for genotyping. Under this conditions, *β-catenin* mutant cells preferentially differentiate into Epi-like fate. In contrast, wild-type cells predominantly differentiate into PrE-like fate (Fig 6C, right). Notably, this approach rescued the defect observed in the distribution of NANOG-expressing cells. In chimeric organoids, high NANOG-expressing cells were no longer found at the periphery but were instead correctly localised near the organoid centroid (Fig 6D, lower panel). Furthermore, NANOG and GATA6 expression levels decrease when comparing all the cells within the chimeric and wild-type ICM organoids. Similar reductions were observed when directly comparing wild-type and mutant cells within the chimeric ICM organoid (S7B Fig).

Chimera experiments also allow determining whether β-catenin acts cell-autonomously or non-cell-autonomously. To test this, we induced GATA6 expression in wild-type cells before mixing them with uninduced *β-catenin* mutant cells, and vice versa (S6 Fig, regime IV and V, respectively). In both scenarios, we observe that PrE-like fate arises exclusively from induced GATA6 expressing cells, with Epi-like cells originating from the uninduced cells. When only *β-catenin* mutant cells were induced (regime V), we also observe a decrease in the percentage of PrE-like cells from *β-catenin* mutant cells. As a control, we generated 3D aggregates using uninduced wild-type and mutant *β-catenin* cells; in this case, we observe no PrE-like cells (S6 Fig, regime VI). Altogether, these findings indicate that PrE-like fate is acquired in a cell-autonomous manner from the induced cells. In other words, induced cells do not promote PrE fate to neighbouring uninduced cells.

In summary, genetic β-catenin inhibition in both 2D or 3D (ICM organoids) *in vitro* models of PrE differentiation leads to a reduced efficiency in PrE-like differentiation, favouring Epi-like fate. The effects are more pronounced in the 2D cell cultures compared to the ICM organoids. This disparity is likely attributable to mechanical factors or differences in the developmental stages represented by the two systems.

### Wnt/β-catenin signalling genetic inhibition influences PrE/Epi fate *in vivo*

Maternally deposited β-catenin in the mouse oocyte is sufficient to successfully complete preimplantation development [31,33]. Hence, to investigate the role of β-catenin during this stage, we generated materno-zygotic (MZ) mutant embryos using the *ZP3*-Cre system [31,46] with *β-catenin^{loxP/-}* mice harbouring the *Ctnnb1^{Tm2Kem}* allele (deletion of exons 2–6) [32]. The control littermates of the MZ embryos are heterozygous for *β-catenin* (see M&M). A previous report indicates that this allele produces a truncated protein version [33]. However, our immunofluorescence analyses of MZ generated embryos with a polyclonal and a monoclonal antibody show no detectable signal (Fig 7, and 8).

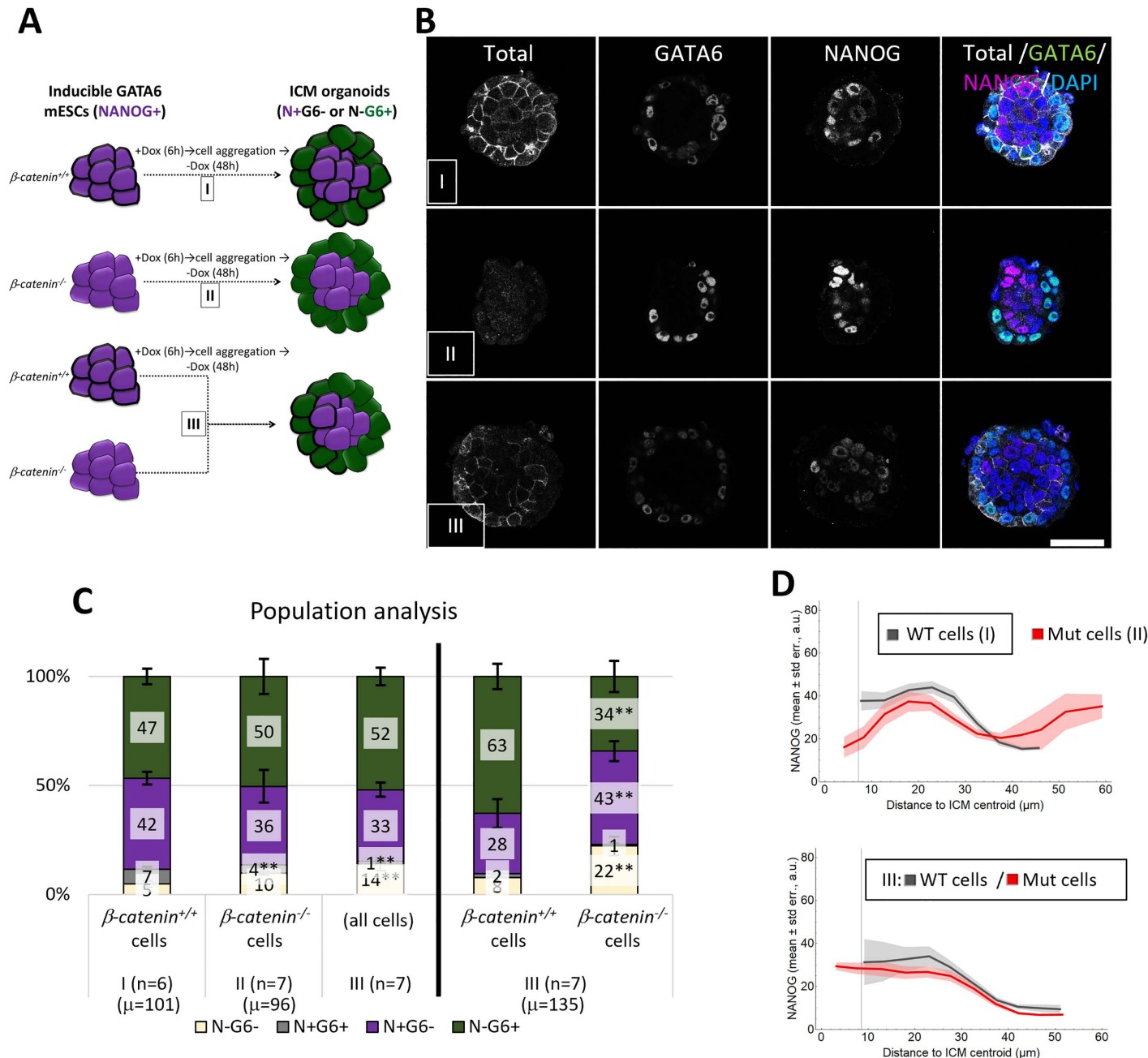

**Fig 6. Genetic Wnt/β-catenin signalling inhibition hinders PrE fate in vitro (3D).** (A) ICM organoids formation regime using *β-catenin +/+* or *β-catenin-/- tet::Gata6*-mCherry mESCs after inducing PrE differentiation (I and II, respectively). Chimeric ICM organoids were formed by mixing equal cell numbers of induced (6h Dox-treated cells) β-catenin +/+ and β-catenin-/- *tet::Gata6*-mCherry cells (III). (B) Representative single confocal images of the ICM organoids generated in each regime and stained with antibodies against total β-catenin (white, used to genotype individual cells), GATA6 (green), and NANOG (magenta), DAPI (blue) was used to stain nuclei. Scale bar: 50 mm. (C) Population analysis of the different ICM organoids cell composition indicating the mean percentage of N-G6-, N + G6 + , N + G6- and N-G6 + cells. Error bars indicate the s.e.m. One-way ANOVA with Bonferroni correction for multiple comparisons was used to compare cell populations´ distribution between whole ICM organoids regimes I and II, and I and III (left). One-tailed unpaired t-test was used to compare cell populations´ distribution between wild-type and mutant cells within regime III ICM chimeric organoids (right). **: $p < 0.05$. n is the number of analysed ICM organoids. μ is the average organoid total cell number. (D) Mean level of NANOG (vertical axis) versus the distance to the ICM centroid (horizontal axis, binned in 5 μm groups) for *β-catenin* $^{+/+}$ (grey) or *β-catenin* $^{-/-}$ (red) cells in ICM organoids of I and II (above) or III (below) regime.

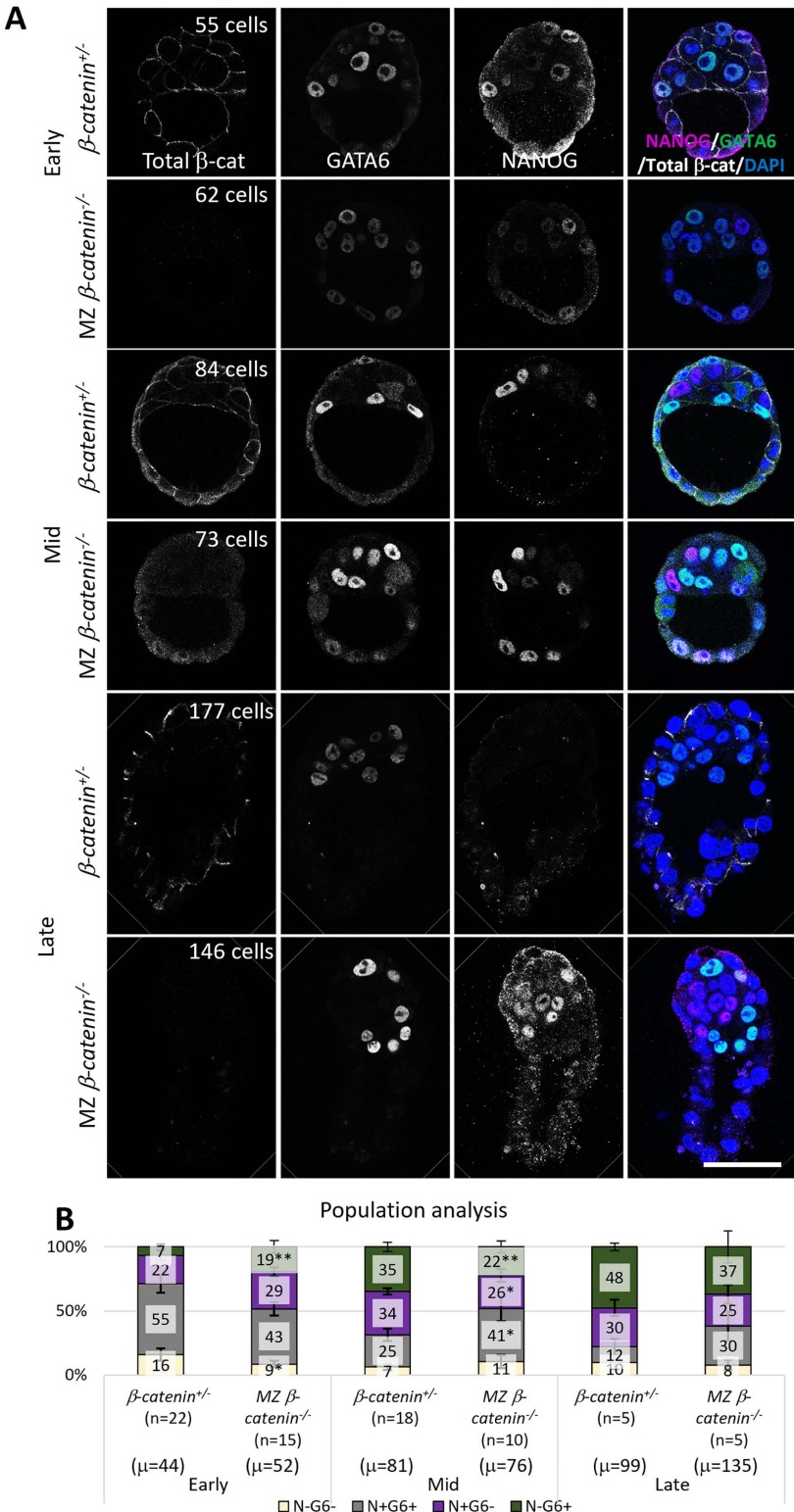

**Fig 7. Genetic Wnt/β-catenin signalling inhibition hinders PrE fate in vivo.** (A) Representative single confocal images of *β*-catenin [+/-] *or* materno-zygotic (MZ) *β*-catenin [-/-] mutant embryos of the indicated stages stained with anti*b*odies against total β-catenin (white), GATA6 (green), and NANOG (magenta), DAPI (*blue*) was used to stain nuclei. Shown embryos from the same developmental stage were immunostained, imaged and processed

together. Scale bar: 50 μm. (B) Population analysis shows the mean percentage of N-G6-, N+G6+, N+G6- and N-G6+ cells. Error bars indicate the s.e.m. One-tailed unpaired t-test was used to compare cell populations´ distribution between β-catenin $^{+/-}$ and materno-zygotic (MZ) β-catenin$^{-/-}$ mutant embryos in each stage. *: p < 0.1, ** < 0.05. n is the number of analysed embryos. μ is the average embryo total cell number.

To assess the role of β-catenin during Epi vs PrE differentiation, MZ *β-catenin* mutant and control heterozygous littermate embryos were stained for NANOG and GATA6, together with an antibody against β-catenin for genotyping (Fig 7 and S8). In early MZ *β-catenin* mutant embryos, there is an increase in PrE differentiation (N-G6+). This is likely due to reduced NANOG expression in N+G6+ cells (S8 Fig). However, in mid and late MZ *β-catenin* mutant embryos, a clear defect in PrE differentiation is observed, along with a higher proportion of cells remaining in an undifferentiated ICM cell state (N+G6+). Furthermore, a lower proportion of Epi cells is found in mid embryos; again, likely due to a decreased NANOG expression levels in Epi (N+G6-) cells. GATA6 levels are also reduced in ICM and PrE cells. In late MZ *β-catenin* mutant embryos, GATA6 levels are elevated in the still undifferentiated ICM cells and NANOG levels in Epi Cells, even though there is no change in the proportion of Epi cells.

In summary, in the absence of *β-catenin*, we observe defects in PrE differentiation: cell fate acquisition seems more efficient in early embryos, but then it stalls in mid embryos with a higher proportion of cells remaining undifferentiated, as well as higher NANOG levels in Epi cells in late embryos.

### Wnt/β-catenin and FGF/MAPK signalling cooperate to induce PrE fate

The main signalling pathway involved in this cell fate acquisition in mouse preimplantation embryos is FGF/MAPK signalling [8,10,15–20]. Hence, we next investigated the relationship between FGF/MAPK and Wnt/β-catenin signalling in this process. *dusp4* is a direct target of FGF/MAPK signalling [47–49] and acts as a negative-feedback regulator by dephosphorylating ERK (reviewed in [50]). During mouse preimplantation development, DUSP4 is accumulated following ERK phosphorylation in presumptive PrE cells [51]. To further investigate the interplay between FGF/MAPK and Wnt/β-catenin signalling, we examined whether the absence of β-catenin influences DUSP4 expression (Fig 8A-8B and S9A). Surprisingly, we observe an increased number of N-DUSP4+ cells in early MZ *β-catenin* mutant embryos (at the expense of N+DUSP4+). In mid embryos, there is an increase of N+DUSP4+ cells (at the expense of N-DUSP4- cells) which clearly show altered DUSP4 expression levels. These results point towards an altered FGF/MAPK signalling in the absence of β-catenin.

To investigate the relationship between FGF/MAPK and Wnt/β-catenin signalling further, we tested whether the defects in PrE differentiation in MZ β-catenin mutant late embryos could be rescued. We treated E3.5 embryos with FGF for 24 hours (Figs 8C-D and S9B). We did observe qualitative changes in the PrE fate upon treatment in the different genetic backgrounds, with FGF treatment rescuing the PrE differentiation defects in the absence of β-catenin. However, the results were not statistically significant. To gain deeper insight, we conducted the reversed experiment: we treated wild-type embryos with FGFRi to inhibit PrE differentiation and tested whether the activating Wnt/β-catenin signalling using Wnt3a could rescue these defects (S9C-D Fig). Under these conditions, activation of Wnt/β-catenin signalling does not rescue the PrE differentiation defects.

Altogether, these results indicate that in the absence of *β-catenin*, DUSP4 accumulates and likely inhibits FGF/MAPK signalling activation. This points towards FGF/MAPK signalling acts downstream or in parallel with Wnt/β-catenin signalling during PrE differentiation.

### β-catenin acts independently of Wnt/β-catenin signalling activation during PrE differentiation

Previous studies using scRNAseq data suggested that Wnt/β-catenin signalling is active in Epi cells of E4.5 embryos [39]. However, the analysis did not cover the differentiation process from E.25 to E4.5 embryos. To investigate Wnt/β-catenin signalling activity throughout ICM differentiation towards PrE and Epi fates, we re-analysed previously published scRNAseq data (Fig 9A) [20,52]. In ICM cells (E3.25 and E3.5 embryos), there is high expression of the Wnt/β-catenin signalling negative regulators *gsk3a* and *apoe*, as well as the positive regulator *fn1*. This pattern persists in E3.5 Epi cells, which becomes even

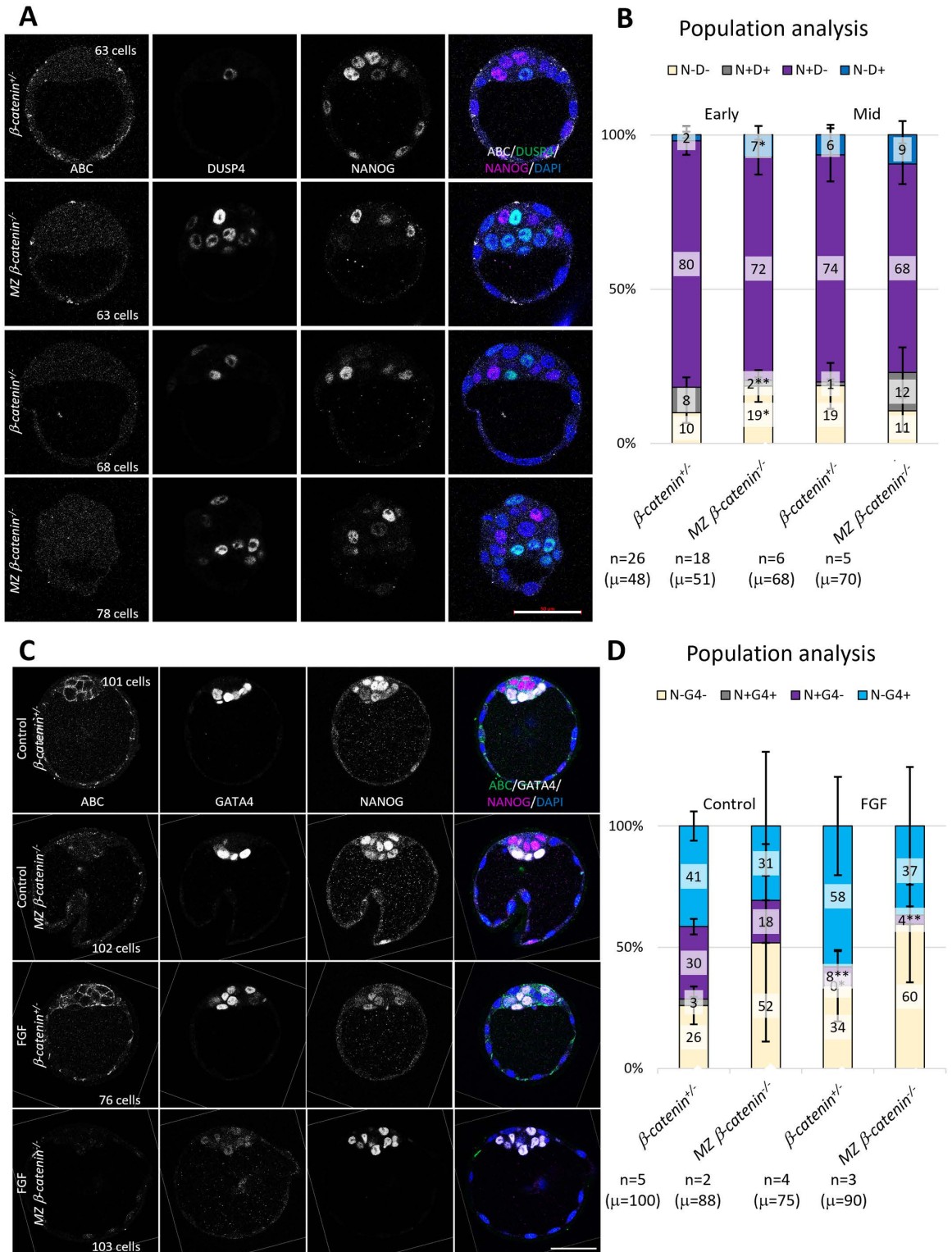

**Fig 8. Wnt/β-catenin signalling cooperates with FGF/MAPK signalling in cell fate decisions in early mouse embryos.** (A) Representative single confocal images of freshly flushed *β-catenin* <sup>+/-</sup> or materno-zygotic (MZ) *β-catenin* <sup>-/-</sup> mutant early embryos and stained with anti*b*odies against active β-catenin (white), DUSP4 (green), NANOG (magenta), and DAPI (*b*lue) was used to stain nuclei. Scale bar: 50 μm. (B) Population analysis indicating the

mean percentage of N-DUSP4-, N+DUSP4 +, N+ DUSP4 – and DUSP4 + cells. Error bars indicate the s.e.m. One-tailed unpaired t-test between β-catenin $^{+/-}$ and materno-zygotic (MZ) β-catenin $^{-/-}$ early or mid embryos. *: p < 0.1; **: p < 0.05. n is the number of analysed embryos. μ is the average embryo total cell number. (C) Representative single confocal images of β-catenin $^{+/-}$ or materno-zygotic (MZ) β-catenin $^{-/-}$ mutant early embryos cultured in control or FGF4-containing medium for 24h and stained with antibodies against active β-catenin (ABC, green), GATA4 (white) and NANOG (magenta), DAPI (blue) was used to stain nuclei. Scale bar: 50 μm. (D) Population analysis indicating the mean percentage of N-G4-, N + G4 +, N + G4- and N-G4 + cells. Error bars indicate the s.e.m. One-way ANOVA with Bonferroni correction for multiple comparisons. The statistical comparison was made against the control treatment (control β-catenin $^{+/-}$ embryos). *: p < 0.1; **: p < 0.05. n is the number of analysed embryos. μ is the average embryo total cell number.

more pronounced in E4.5 Epi cells. In these cells, another Wnt positive regulator, *lef1*, is also highly expressed. In PrE cells, both *gsk3a* and *apoe* are expressed (in E3.5 and E4.5 embryos); the key differences between PrE and Epi cells lie in the high expression of *dkk1* and the lower expression of *fn1* in PrE cells, the latter specially in the late PrE cells. As previously suggested [39], these results point towards Wnt/β-catenin signalling being activated in Epi cells but inhibited in PrE cells.

We also analysed whether *β-catenin* (*ctnnb1*) expression is differentially regulated during early development (S10A Fig). As previously reported, we observe that *β-catenin* is expressed at similar levels across all stages and fates, validating the need of materno-zygotic mutant embryos to investigate its role [31,33]. Interestingly, the only statistically significant difference in expression levels occurs between ICM and PrE cells in E3.5 embryos, suggesting a role for β-catenin in PrE cells. For comparison, the expression levels of other fate-specific genes are also shown (S10B Fig). During this analysis, we identified two genes which exhibit clear upregulation in PrE cells, namely *sparc* and *aqp8*. While these genes have been previously studied in the context of mouse preimplantation development, no research has been conducted to investigate points towards a role in PrE vs Epi differentiation [53–55].

The scRNAseq data analyses are at odds with the results shown in this study, which suggest that β-catenin activates PrE fate. Our previous work showed a role for β-catenin in the maintenance of pluripotency by regulating pluripotency markers via protein complexes stabilities, rather than promoting their transcriptional activation via Wnt/β-catenin signalling activation [27,28]. Within this context, we hypothesised that β-catenin might exert a similar function. To test this hypothesis, we measured the stability of NANOG, GATA6 and GATA4 using quantitative immunofluorescence analysis after promoting Epi- or PrE-like fate in the 2D *β-catenin* wild-type or mutant cells while blocking translation (adding cycloheximide) for 0, 2, 4 or 6h to the *tet::Gata6*-mCherry mESCs (Fig 9B-E and S11). To estimate NANOG, GATA6 and GATA4 stability, we fitted a first-order decay model using the measured protein levels (dashed lines in Fig 9C-E). Upon levels normalization under both genetic backgrounds, we observed no changes in NANOG stability (Fig 9C). However, both GATA6 and GATA4 stabilities are significantly affected by the absence of β-catenin (Fig 9D-E).

This suggests that β-catenin contributes to PrE differentiation post-transcriptionally by stabilising GATA6 and GATA4.

## Discussion

Previous studies did not identify a role for Wnt/β-catenin signalling during mouse preimplantation development and Epi versus PrE differentiation [30–35]. This was likely due to the use of qualitative rather than quantitative approaches. We have previously proposed a role in the process based on mESCs studies and given the embryonic origin of these cells [26–28]. Furthermore, other studies have hinted at a potential role: in the *in vitro* PrE-like differentiation model, the addition of Chi to the culture enhances PrE-like differentiation and increases the number of PrE cells in the embryo [37,38]. Furthermore, Wnt, in combination with ActA, induces PrE in naïve pluripotent cells [36]. Here, we quantitatively analyse the influence of Wnt/β-catenin signalling during mouse Epi and PrE differentiation using 2D and 3D *in vitro* models, as well as *in vivo* models. To this end, we apply chemical and genetic activation or inhibition of the pathway. In summary, we find that activation of the pathway enhances PrE differentiation, whereas its inhibition stalls it. For this activity Wnt/β-catenin cooperates with FGF/MAPK signalling.

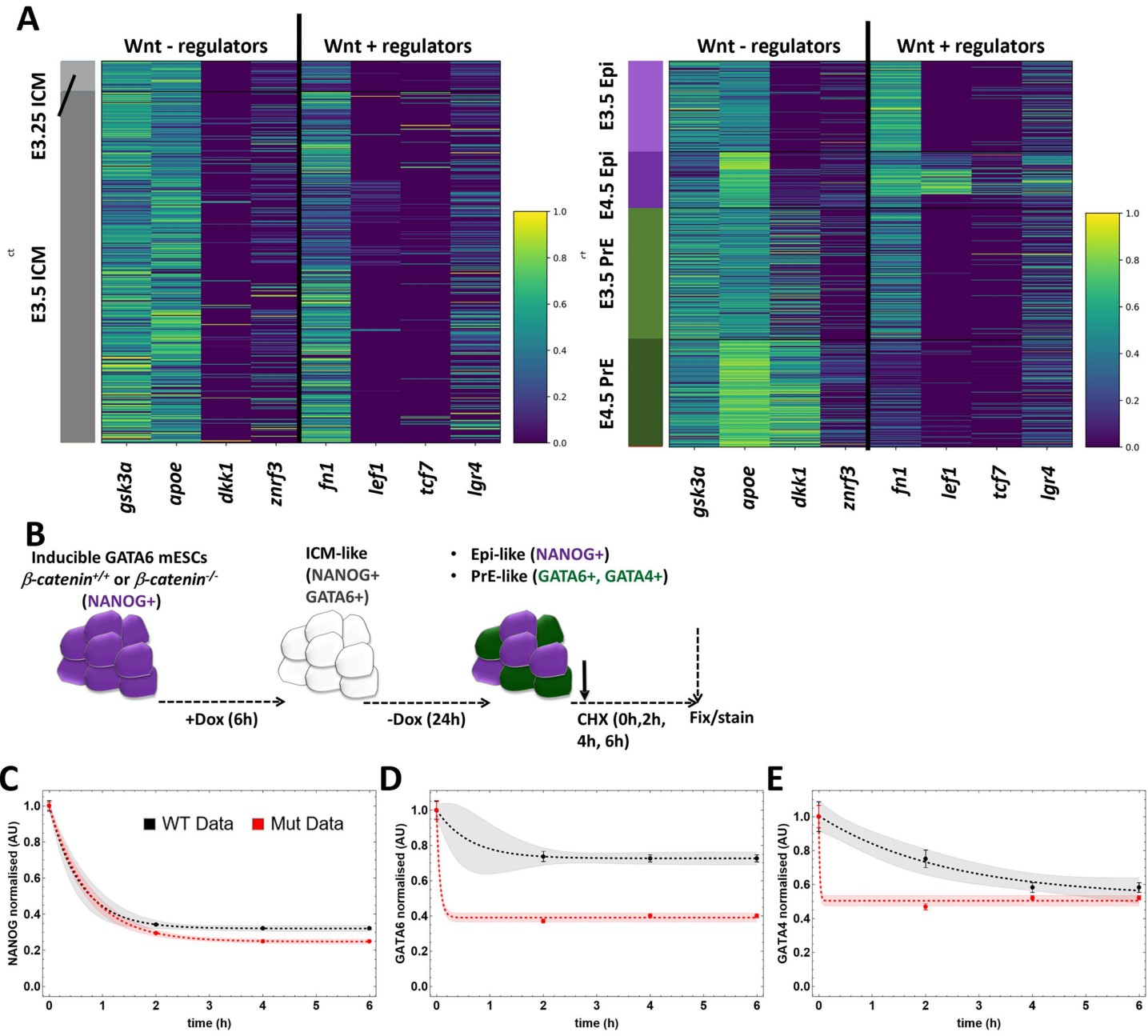

**Fig 9. β-catenin acts independently of Wnt/β-catenin signalling activation during PrE differentiation via GATA6 and GATA4 turnover regulation. (A)** scRNAseq expression analyses of Wnt/β-catenin signalling positive (*fn1, lef1, tcf7* and *lgr4*) and negative regulators (*gsk3a, apoe, dkk1* and *znrf3*) in ICM cells from E3.25 embryos (left), and Epi and PrE lineages in E3.5 and 4.5 embryos (right). **(B)** Treatment regime of wild-type and mutant β-catenin mutant *tet::Gata6*-mCherry mESCs to measure NANOG, GATA6 and GATA4 half-lives. **(C-E)** Average levels and standard error of NANOG (C), GATA6 (D), and GATA4 (E) normalised to levels at 0h. Wild-type cells levels are in black, and mutant cells levels are in red. Dashed lines correspond to the *b*est fit exponential decay model using the experimental data from wild-type and mutant cells, respectively. Shading shows the 90% confidence bands.

In this study we use three biological systems (2D cell culture and ICM organoids), and embryos to manipulate Wnt/β-catenin signalling and investigate its role during cell fate allocation. Our results allow us to propose that it promotes PrE differentiation. The different systems allow us to address various aspects of the process, as recently shown with FGF/MAPK response in mESCs and embryos [56]. The 2D system is based on mESCs genetically "forced" to differentiate into PrE-like cells via GATA6 overexpression. Consequently, the results we observe represent a mixture of the effects that Wnt/β-catenin signalling has on pluripotency and differentiation [27,28,57]. Increasing β-catenin levels (via the addition of Chi) inhibits pluripotency exit, resulting in the observation of more ICM-like cells (due to ectopic GATA6 expression). In cells primed for differentiation, higher β-catenin promotes pluripotency exit, and, under this condition, we observe more PrE-like cells. For the opposite situation, we have two scenarios: one with lower β-catenin levels (XAV treatment) and the other with no β-catenin at all (mutant cells). In both cases we observe a reduction in PrE-like differentiation efficiency and an increase in cells expressing neither Epi (NANOG), nor PrE markers (SOX17, GATA4). These later cells might reflect cells differentiating into more advanced embryonic fates [9]. The key difference between these two scenarios lies in the NANOG expressing cells (Epi-like cells): under XAV treatment, there is sufficient β-catenin to support the presence of these cells; however, in the absence of β-catenin, we find fewer Epi-like cells, as NANOG decreases its stability in mESCs or maybe reflecting a more advanced Epi state [9,27].

In the ICM organoids, we force cells to differentiate while forming a 3D structure resembling the ICM [45]. In this system, we only observe a defect in PrE-like differentiation when mixing *β-catenin* wild-type and mutant cells to generate a chimeric ICM organoid. In the absence of β-catenin, cells preferentially differentiate into Epi-like cells, with wild-type cells comprising most PrE-like cells. ICM organoids composed entirely of mutant cells do not exhibit any differentiation defect, but show altered cell distribution, with Epi-like cells reaching the periphery of the organoid. This may reflect defects in cell adhesion within mutant PrE-like cells. Interestingly, this is the only adhesion-related defect identified in our study, despite the role of β-catenin at the adherens junctions [26,33,58,59].

Finally, the results obtained from the *in vivo* system (i. e., the embryo) allow us to hypothesise how Wnt/β-catenin signalling is involved in mouse preimplantation cell fate decisions at different stages. We observe a clear defect in materno-zygotic *β-catenin* mutant embryos in PrE differentiation, alongside an increase in undifferentiated ICM cells co-expressing NANOG and GATA6 (in mid and, only qualitatively, in late embryos), which has only partially been observed in the *in vitro* models. This leads us to propose that, in the absence of β-catenin, the cell fate acquisition is delayed. In the *in vitro* systems, cells do have time to process new differentiation directions and transit through different states (naïve, formative, primed). However, the mouse embryo has a finite time to progress through development till birth. Thus, it is unsurprising that the effects of Wnt/β-catenin vary, given its role in cell state transitions in mouse [48,60,61].

The mechanism by which Wnt/β-catenin signalling contributes to the transition from ICM to PrE/Epi fate is likely complex and context dependent as suggested for FGF/MAPK signalling [56]. In our study, we observe at least two ways by which Wnt/β-catenin signalling is involved. The first one involves the crosstalk with FGF/MAPK signalling. Our results indicate that Wnt/β-catenin signalling acts downstream of or in parallel with FGF. Interestingly, mouse preimplantation development is not the only system where these two pathways interact. Both in basal cells of the trachea and in neuromesodermal progenitors these two pathways are active [62,63]. The second mechanism is via GATA6 and GATA4 protein stability. Given the autoactivation of *gata6* expression during PrE differentiation by binding to its own promoter but also to the *gata4* promoter [64,65], the reduced GATA6 stability in the absence of β-catenin explains the observed defects (lower GATA6/4 levels and decreased PrE differentiation). Surprisingly, just 2h after stopping translation (CHX treatment), both GATA6 and GATA4 levels decrease significantly, indicating that their dynamics are notably fast in the absence of β-catenin. These two mechanisms are not independent and they likely coexist in the process.

During Epi versus PrE differentiation from E3.5 embryos' ICM cells, the expression profiles indicate a differential regulation of Wnt/β-catenin signalling. Specifically, in PrE cells at both E3.5 and E4.5, negative regulators such as *gsk3a* and *apoe* are expressed, while *fn1* (activator) shows lower expression, especially in late PrE cells. Conversely, E3.5 Epi

cells maintain a persistent expression pattern of the positive regulators *fn1* and *lef1*, which becomes more pronounced in E4.5 Epi cells. These findings suggest that Wnt/β-catenin signalling is activated in Epi cells but inhibited in PrE cells. This supports the premise that β-catenin plays a role in PrE fate acquisition, potentially acting independently of the canonical Wnt/β-catenin signalling activation by regulating GATA6 and GATA4 turnover. While our current results may not show clear defects in Epi cells, the activation of the pathway during Epi fate in late blastocysts, as hinted by scRNAseq analyses, appears to be at odds with direct experimental evidence suggesting β-catenin activates PrE fate. This discrepancy could be due to scRNAseq results capturing a different dimension or a snapshot that doesn't fully reflect the dynamic, post-translational, or indirect transcriptional influences observed in functional experiments.

Interesting findings emerge from the analysis of DUSP4. *dusp4* expression and protein levels oscillate during somitogenesis [49] and act as a negative-feedback regulator of FGF/MAPK signalling by dephosphorylating ERK (reviewed in [50]). This regulation also occurs during PrE differentiation [21,51,66]. Oscillations in FGF/MAPK signalling have been observed in mouse embryonic stem cells [66], in cells exiting pluripotency [67], and in mouse blastocysts [21]. Specifically, studies on pluripotency exit demonstrate that, rather than MAPK dynamics, it is the cumulative ERK activity experienced by the cells that determines the outcome [67]. Additionally, FGF promotes more cells entering an oscillatory regime, which would safeguard cells from spurious signal activation [66]. While no *dusp4* oscillations have been reported during mouse preimplantation development, the presence of distinct high- and low-*dusp4*-expressing cell populations in PrE cells of E3.5 embryos [51] suggests a similar behaviour in this system. In our study, we observe different DUSP4 levels as development progresses as published before ([51], see S9A Fig), which are altered in β-catenin mutant embryos. It is tempting to propose that Wnt/β-catenin might regulate DUSP4 activity and, in turn, influence MAPK signalling oscillations. Indeed, in the early MZ *β-catenin* mutant embryos, we observe an increase in DUSP4+ cells and more PrE cells (Fig 8B); in the mid embryos, a new population of cells emerges in which both NANOG and DUSP4 are co-expressed (Fig 8B).

Overall, we can propose the working model shown in Fig 10, which combines our results (blue lines) with previous finding (black lines). In wild-type ICM cells FGF/MAPK signalling oscillates (Fig 10A [21,66–68]). High FGF/MAPK signalling (upper part in Fig 10A) activates GATA6 and inhibits NANOG (via phosphorylation through ERK) [65,69]; GATA6 reinforces PrE fate by promoting its own expression [64,65], while inhibiting Epi fate by repressing *nanog* [64,70]. FGF/MAPK signalling promotes *dusp4* expression [47–49,51], which acts as a negative-feedback regulator by dephosphorylating ERK during PrE differentiation [21,51,66]. This negative feedback would result into low FGF/MAPK signalling (lower part in Fig 10A); under these conditions, NANOG phosphorylation decreases, leading to an increase NANOG levels, inhibition of *gata6* expression [71], and progression towards Epi fate. In the absence of β-catenin (Fig 10B), GATA6 stability decreases and DUSP4 is accumulated. This would result in reduced FGF/MAPK signalling activation, decreased ERK activity, further diminished GATA6 activity, increased NANOG activity and, consequently, reduced PrE fate.

## Materials and methods

### Mice and embryos

Mouse work was conducted following National and European regulations (RD 1201/2005, Law 32/2007, EU Directive 2010/63/EU), were approved by the Animal Ethics Committee of the ULPGC and were authorised by the competent authority of the Canary Islands Government (reference number: OEBA-ULPGC 08/2018). Initial mouse work in this project was approved by the University of Bath Animal Welfare and Ethical Review Body (AWERB) and undertaken under a UK Home Office license (PPL 30/3219) in accordance with the Animals (Scientific Procedures) Act incorporating EU Directive 2010/63/EU. Mice were maintained under a 14h light/10h dark cycle with food and water supplied *ad libitum*. Females were sacrificed by neck dislocation; all efforts were made to minimize suffering, including environmental enrichment during cage maintenance to reduce stress levels.

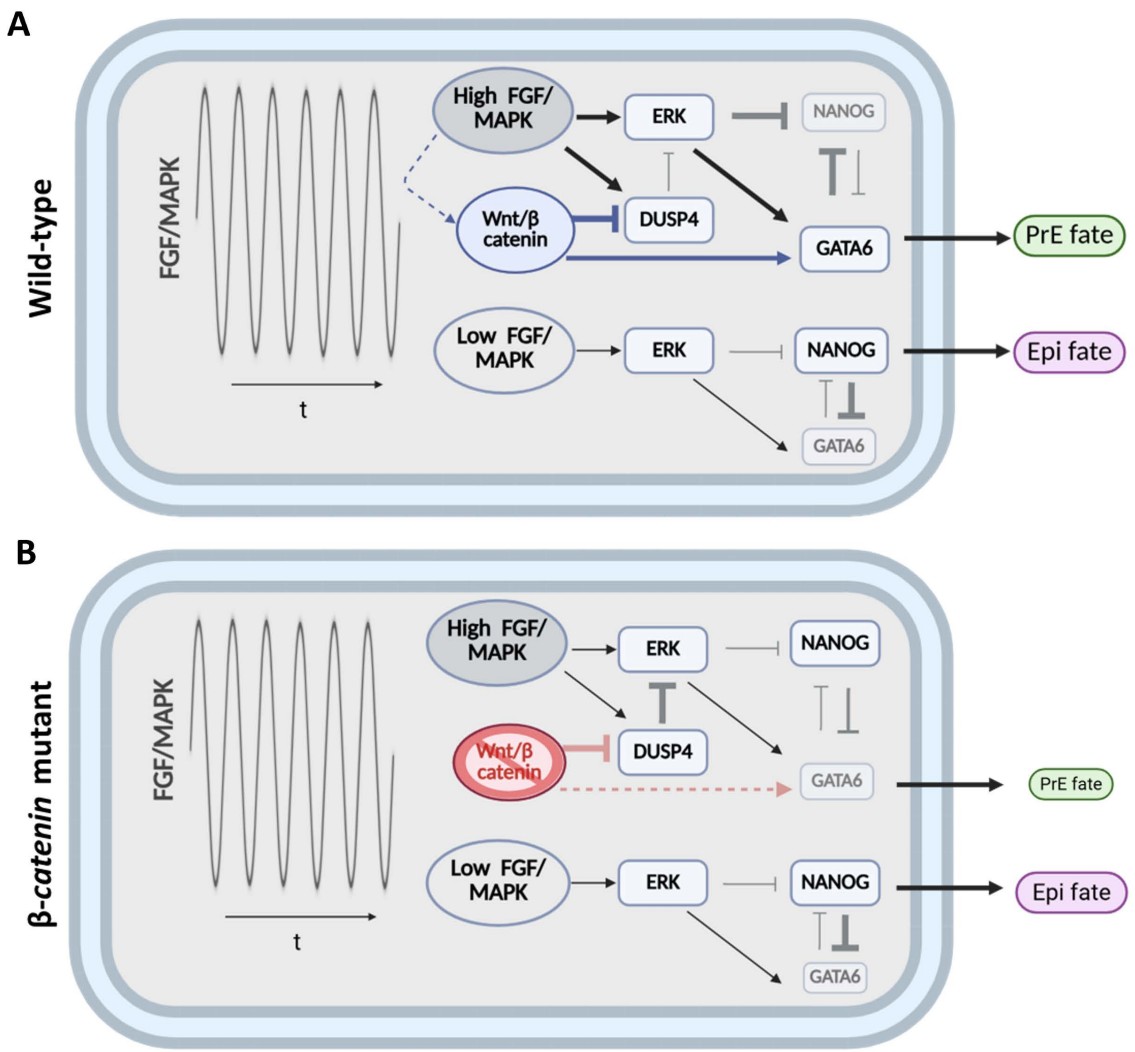

**Fig 10. Working model for the role of Wnt/β-catenin during early mouse preimplantation embryo cell fate decision.** Top scheme represents the wild-type situation with previously described relationships in black. The effects of the pathway on the process identified in this study are in blue. The bottom scheme represents how cell fate decision is affected in the absence of *β-catenin* (red lines), increasing or decreasing line weights or letter size accordingly. For simplicity transcriptional and posttranslational effects are not depicted differently. See main text for details.

Mouse strains: CD1, and Swiss as wild-type strains. *Zp3*-Cre (C57BL/6-Tg(Zp3-cre)93Knw/J, JAX stock Nº: 003651) [72]; *Sox2*-Cre (B6.Cg-*Edil3*^Tg(Sox2-cre)1Amc^/J, JAX stock Nº: 008454) [73]; β-catenin^loxP/loxP^ (B6.129-*Ctnnb1*^tm2Kem^/KnwJ, JAX stock Nº: 004152) [32]; β-catenin ^+/-^ (generated in house by mating β-catenin^loxP/loxP^ with *Sox2*-Cre). Generation of materno-zygotic β-catenin mutant embryos was done as in [46].

Mouse genotyping primers were: β-catFLOXRM41 (AAGGTAGAGTGATGAAAGTTGTT), β-catFLOXRM42 (CAC-CATGTCCTCTGTCTATTC), β-cat^+/-^ RM68 (AATCACAGGGACTTCCATACCAG), β-cat^+/-^ RM69 (GCCCAGCCTTAGC-CCAACT), Cre-oIMR 1084 (GCGGTCTGGCAGTAAAAACTATC), Cre-oIMR 1085 (GTGAAACAGCATTGCTGTCACTT), Cre-oIMR 7338 (CTAGGCCACAGAATTGAAAGATCT) and Cre-oIMR 7339 (GTAGGTGGAAATTCTAGCATCATCC).

Embryos used for this study were obtained via natural mating and flushed using M2 medium (Embryomax®; Millipore, Ref. MR-015-D). For culture, embryos were flushed at E3.5 and cultured in KSOM (Millipore, Ref. MR-121-D) at 37°C and

5% $CO_2$ [37] using organ culture dishes with 500μl medium with or without small molecules as in [37]. PBS was added to the exterior part of the dish to avoid evaporation. To avoid litter effects, uteri from females from the same genotype were collected and flushed in the same plate. Around 5–6 embryos per female were obtained and randomly cultured in control or experimental conditions. All embryos from the same genotype and experimental condition were cultured together. Small molecules used were: Chiron9901 (Eurodiagnostico HY-10182, 3 μM) [37], Wnt-3a (R&D, 1324-WN-002, 200ng/ml) [35,40], XAV939 (R&D, 3740, 1μM) [28,43], FGF2 (R&D, 233-FB-025, 500ng) [5], FGFRi (AZD4547, Abcam, 1 μM) [5] and PD035901 (TOCRIS, 4192, 1μM) [5,37] were added one hour before starting the *ex vivo* culture to equilibrate the medium in the incubator.

## Mouse embryonic stem cell (mESC) lines

*tet::Gata6*-mCherry mESCs were kindly provided by Christian Schröter [38]. A truncated β-catenin GATA6 inducible cell line was obtained by CRISPR mutation using ready-made plasmids from Santa Cruz (sc-419477) formed by three different gRNA (one: ATGAGCAGCGTCAAACTGCG; two: AGCTACTTGCTCTTGCGTGA; three: AAAATGGCAGT-GCGCCTAGC). *tet::Gata6*-mCherry mESCs were transfected chemically using the transfection reagent (Santa Cruz, sc-395739) and transfection medium (Santa Cruz, sc-108062). Transfected mESCs were sorted based on GFP expression using a FACS Aria (BD Biosciences) 8 hours after transfection. The GFP fluorescence threshold was achieved using non-transfected cells as control. GFP positive cells were grown on iMEFs coated plates. Individually picked clones were screened using genomic PCR with primers designed for the 3 gRNA included in the transfected plasmid (F: CTGGCAG-CAGCAGTCTTACT, R: GCACCGTACTGTACACACAGA). Sequencing data indicated that the F1 *β-catenin* mutant clone was not pure and was mixed with wild-type cells. Hence, unless otherwise indicated in the figures, only results from the C5 clone are shown in comparison with a C4 untransfected clone (wild-type control cells).

mESCs culture was performed as previously described [27].

## ICM organoids

ICM organoids were generated as previously published, using 50 cells as starting population in ultra-low 96-well round bottom plates [45]. To generate the chimeric ICM organoids, 25 cells of each genotype and/or treatment were used to obtain a total of 50 cells, used as starting number. Total-β-catenin staining was used to manually genotype individual cells. Total cell number in individual organoids range between 73 and 218 cells, in none of them antibody penetration issues were detected.

## mESCs, ICM organoids and embryos immunostaining, and imaging

Primary antibodies include: NANOG (1:200, eBIOSCIENCE, 14-5761-80), OCT3/4 (1:200, SANTA CRUZ, SC5279), GATA6 (1:200, R&D, AF1700), SOX17 (1:200, R&D, AF1924), GATA4 (1:200, SANTA CRUZ, SC9053), Total β-catenin (1:500–1:1000, SIGMA, C2206), active β-catenin (1:300, MILLIPORE, 05–665), DUSP4 (1:100, Abcam, ab216576). Nuclei were visualized using Hoechst (Invitrogen, H3570) or DAPI (Invitrogen, D1306).

To visualise DUSP4, a tyramide signal amplification (TSA) reaction was carried out according to manufacturer's instructions (Invitrogen, B40943) and published protocol [51].

mESCs, ICM organoids and embryos were immunostained, and confocal imaged as previously described [13,27,45]. To determine protein stabilities, mESCs were induced to differentiate to Epi- or PrE-like as previously published by adding Dox for 6h [38]. 24h after Dox removal, 40 μM cycloheximide was added to inhibit translation for 0h (control), 2h, 4h or 6h as in [27].

Confocal images were acquired using a Zeiss LSM-510-META with a Plan-Apochromat 63x/1.4 Oil Ph3 objective with 0.7x zoom, LSM-880+Airyscan with a 40x/1.3 Oil DIC UV-IR M27 objective or Zeiss LSM700 and a Plan-Apochromat 40x/1.3 Oil DIC (UV) VIS-IR M27 objective laser scanning confocal microscopes. All images in each imaging session were

obtained using the sequential scanning mode, with the same conditions of laser intensity, gain, and pinhole, and were processed in exactly the same way, see [12].

For the cells experiments, randomly selected 5–10 fields were used for imaging and posterior analyses. For ICM organoids, all obtained samples were imaged. For cultured mouse embryo experiments, E3.5 embryos (32–64 total cell number) were flushed and only those embryos with over 75 total cell number after the 24h culture were considered in the analysis.

## Image and data analyses

Images and data were analysed as previously described [12,13,27,45] using MINS [44]. For more details on the pipeline for the correction of the fate markers levels variability between experiments, see [12].

Exponential decay to estimate protein stability was modelled using Mathematica 11.0.

To quantify membrane β-catenin, ARIVIS software was used to help with visualization. ImageJ was used for quantification by drawing a line across the membrane between the indicated cells. Cells to quantify were manually selected to include membranes between the different cell fates (N+G6+, N+G6- and or N-G6+).

Data analyses were performed [12,13] using Paleontological Statistics (PAST) [74], GraphPad Prism, Matlab R2017b, and Mathematica 11.0.

## Statistical analyses

For the comparison of the means between two experimental set ups unpaired t-test was used. In those cases where variances where statistically significant different t (as determined by the F test), the t-test was performed with the Welch correction. When more than two means were to be compared, one-way ANOVA test with Bonferroni correction was used. These tests were done in GraphPad Prism 8. Z-test was applied to compare cell populations; the requirements for this test are: random sample and categorical data; this test was performed in https://www.socscistatistics.com/tests/ztest/. One-tailed tests offer greater statistical power to detect an effect in the chosen direction but are limited to that specific direction. Two-tailed tests are more conservative and can detect differences in either direction but require a larger sample size or have lower power to detect an effect in any one direction. Hence, one-tailed hypothesis were applied to those obtained from embryos and organoids, while two-tailed hypothesis were applied to experiments undertaken in cells. See S1 Table.

## Western blotting

Western blot was performed as previously published [28]. Briefly, cells were lysated with RIPA buffer supplemented with a protease inhibitor cocktail (PPC1010, Sigma-Aldrich), centrifuged 10000g for 10min, quantify using a BCA assay and preserved frozen at −80°C. 120mg of total protein were loaded into a polyacrylamide gel. Anti GAPDH (1:5000, Sigma, 9545) was used as loading control.

## scRNAseq data analyses

Mouse single-cell RNA-seq data is already published and may be accessed at https://brickman-preimplantation.streamlit.app/Download (file: 01_mouse_reprocessed.h5ad) [52]. Plots were performed with Python (v3.12.3) using specific libraries such as Scanpy (v1.10.4), Seaborn (v0.13.2), Matplotlib (v3.8.4), NumPy (v1.26.4), Pandas (v1.5.3), and statannotations (0v.7.1). Jupyter code is available at Github: https://github.com/kuaiat/Jupyter.

## Use of artificial intelligence tools

We utilized Copilot to assist in the refinement of the manuscript and ChatGPT for Python code assistance and debugging.

## Acknowledgment

Animal work at the ULPGC was done at the SABiA (Servicio Científico Servicio de Animalario y Bienestar Animal).

## Supporting information

**S1 Fig. Active β-catenin localises in cell membranes in mouse preimplantation embryos.** Representative single confocal images of early, and late mouse blastocysts stained with antibodies against GATA6 (green), active β-catenin (red), OCT4 (magenta), DAPI (blue) was used to stain nuclei. Scale bar: 50 μm.
(TIF)

**S2 Fig. Chemical Wnt/β-catenin signalling modulation influences PrE fate in vitro. (A)** Representative single confocal images of *tet::Gata6*-mCherry cells cultured under the indicated regimes in Fig. 3A and stained with an antibody against total β-catenin to reveal the inhibitors' effect. Scale bar: 50 μm. **(B)** Scatter plots showing the quantitative single cell analyses of the indicated markers in the indicated cell types cultured in the different regimes shown in Fig. 3. Red line shows the mean expression level. One-way ANOVA test with Bonferroni correction for multiple comparisons, *: $p < 0.1$, **: $p < 0.05$. The statistical comparison was made against the control regime. n is the number of analysed cells.
(TIF)

**S3 Fig. Chemical Wnt/β-catenin signalling activation promotes PrE fate in vivo.** Scatter plots showing the quantitative single cell analyses of the indicated markers in the indicated cell types shown in Fig. 4. Red line shows the mean expression level. One-tailed unpaired t-test, *: $p < 0.1$, **: $p < 0.05$. n is the number of analysed cells.
(TIF)

**S4 Fig. β-catenin -/- tet::Gata6-mCherry mESCs characterisation. (A)** NCBI scheme (*Mus musculus* strain C57BL/6J chromosome 9, GRCm39 gi|1877089960|ref|NC_000075.7|) of *ctnnb1* gene showing the position of gRNA´s used to generate the mutant cell line (red). Primers´ position used to screen the generated clones (blue) are also indicated. **(B)** Schematic diagram of the β-catenin protein structure with functional domains. The gRNAs positions are indicated. **(C)** β-catenin protein sequence with gRNAs shown in red. **(D)** PCR results using the primers shown in (A) using control *β*-catenin$^{+/+}$ (C4 clone) and *β*-catenin $^{-/-}$ (C5, left, and F1, right, clones) cells´ genomic DNA. **(E)** Population analyses show the mean percentage of N-G6-, N+G6+, N+G6- and N-G6+ (left) or N-G4-, N+G4+, N+G4- and N-G4+ (right) in control *β*-catenin $^{+/+}$ (C4 clone) and *β*-catenin $^{-/-}$ (F1 clone) cells. Two tailed-Z-test, **: $p < 0.05$. **(F)** *ctnnb1* nucleotide coding sequence with gRNAs shown in red. The different coding exons are shown in alternating colours (black and green). The blue *b*ox shows the deleted region in C5 clone. **(G)** Representative single confocal images of *β*-catenin $^{+/+}$ *tet::Gata6*-mCherry (*a*bove) and *β*-catenin $^{-/-}$ *tet::Gata6*-mCherry (C5 clone) mESCs with anti*b*odies against NANOG (green), OCT4 (red) and total β-catenin (white), DAPI (*b*lue) was used to stain nuclei. Scale bar: 50 μm. **(H)** Western blot analysis of *β*-catenin $^{+/+}$ and *β*-catenin $^{-/-}$ (C5 clone) mESCs total protein extracts using total β-catenin antibody. An antibody against GAPDH was used as loading control.
(TIF)

**S5 Fig. Genetic Wnt/β-catenin signalling inhibition hinders PrE fate in vitro (2D).** Scatter plots showing the quantitative single cell analyses of the indicated markers in the indicated cell types shown in Fig. 5. Red line shows the mean expression level. Two-tailed unpaired t-test, *: $p < 0.1$, **: $p < 0.05$. n is the number of analysed cells.
(TIF)

**S6 Fig. Genetic Wnt/β-catenin signalling inhibition hinders PrE fate in vitro (3D). (A)** Complete ICM organoids formation regime using *β*-catenin $^{+/+}$ or *β*-catenin $^{-/-}$ of *tet::Gata6*-mCherry mESCs after inducing PrE differentiation (I and II, respectively). Chimeric ICM organoids were formed by mixing induced *β*-catenin $^{+/+}$ with uninduced *β*-catenin

*-/-* tet::Gata6-mCherry cells (III), or induced β-catenin *-/-* with uninduced β-catenin *+/+* tet::Gata6-mCherry cells (IV). Chimeric aggregates with uninduced β-catenin *+/+* and β-catenin *-/-* of tet::Gata6-mCherry cells were also generated as control. **(B)** Representative single confocal images of the ICM organoids generated in each regime and stained with antibodies against total β-catenin (white), GATA6 (green) and NANOG (magenta), DAPI (blue) was used to stain nuclei. Scale bar: 50 μm. **(C)** Population analysis of the different ICM organoids cell composition indicating the mean percentage of N-G6-, N+G6+, N+G6- and N-G6+ cells. Error bars indicate the s.e.m. One-way ANOVA test with Bonferroni correction for multiple comparisons was used to compare cell populations´ distribution between whole ICM organoids regimes I and II, and I and III (left). One-tailed unpaired t-test was used to compare cell populations´ distribution between wild-type and mutant cells within each ICM chimeric organoids (regimes III-VI, right). **: $p < 0.05$. n is the number of analysed ICM organoids. μ is the average organoid total cell number.
(TIF)

**S7 Fig. Genetic Wnt/β-catenin signalling inhibition hinders PrE fate in vitro (3D). (A)** Scatter plot showing the total cell number in each individual analysed ICM organoids shown in Fig. 6. Red line shows the mean cell number. One-way ANOVA test with Bonferroni correction for multiple comparisons between I and II or III, and between β-catenin *+/+* and β-catenin *-/-* cells in III does not show a statistically significant difference at a significance level of $p < 0.05$. n is the number of analysed ICM organoids. **(B)** Scatter plots showing the quantitative single cell analyses of the fate markers in the indicated cell types shown in Fig. 5. Red line shows the mean expression level. One-way ANOVA test with Bonferroni correction between I and II or III (considering all cells), and between WT and mutant cells within III, **: $p < 0.05$. n is the number of analysed cells. **(C)** Mean level of GATA6 (vertical axis) versus the distance to the ICM centroid (horizontal axis, binned in 5 μm groups) for ICM organoid β-catenin *+/+* (grey) or β-catenin *-/-* (red) cells in ICM organoids of I and II (above) or III (below) regime.
(TIF)

**S8 Fig. Genetic Wnt/β-catenin signalling inhibition hinders PrE fate in vivo.** Scatter plots showing the quantitative single cell analysis of the fate markers in the indicated cell types shown in Fig. 7B. Red line shows the mean expression level. One-tailed unpaired t-test was used to compare marker levels in cells within the same stage, *: $p < 0.1$, **: $p < 0.05$. n is the number of analysed cells.
(TIF)

**S9 Fig. Wnt/β-catenin signalling cooperates with FGF/MAPK signalling in cell fate decisions in early mouse embryos. (A)** Scatter plots showing the quantitative single cell analysis of the indicated markers in the indicated populations in cells shown in Fig. 8B. One-tailed unpaired t-test comparing marker levels in the indicated cell populations within each stage, *: $p < 0.1$. n is the number of analysed cells. **(B)** Scatter plots showing the quantitative single cell analysis of the fate markers in the indicated populations in cells shown in Fig. 8D. Red line shows the mean expression level. One-way ANOVA test with Bonferroni correction for multiple comparisons was used to compare marker levels in all conditions, **: $p < 0.05$. n is the number of analysed cells. **(C)** Population analysis indicating the mean percentage of N-G4-, N+G4+, N+G4- and N-G4+ cells in wild-type embryos cultured in control, FGFRi, Wnt3a or FGFRi+Wnt3a-containing medium for 24h. Error bars indicate the s.e.m. One-way ANOVA test with Bonferroni correction between control and all conditions, **: $p < 0.05$. n is the number of analysed cells. μ is the average embryo total cell number. **(D)** Scatter plots showing the quantitative single cell analysis of the fate markers in the indicated populations in cells shown in C. Red line shows the mean expression level. One-way ANOVA test with Bonferroni correction between all conditions, **: $p < 0.05$. n is the number of analysed cells.
(TIF)

**S10 Fig. Single cell expression analyses of relevant genes during mouse preimplantation development. (A)** ctnnb1 expression levels at the indicated stages and cell types. Notice the statistically significant higher expression in PrE cells vs ICM cells in E3.5 embryos. **(B)** dusp4, gata6, gata4 and nanog levels. Notice dusp4 higher expression in PrE cells in E3.5

embryos, with clearly two populations present (one with high expression and another with low expression). The presence of a high and low expression population in these genes can only be found in *gata4* and *nanog* in the same cell type and stage. **(C)** *sparc* and *apq8* expression levels. These genes are the highest expressed genes in PrE cells, especially in E3.5 and 4.5 embryos. t-test independent samples with Bonferroni correction. * p<= 0.05. For simplicity, only relevant statistical comparisons are shown.
(S10_Fig.TIF)

**S11 Fig. β-catenin absence accelerates GATA6 and GATA4 turnover. (A)** Population analysis showing the mean percentage of N-G6-G4- (yellow), ICM-like cells (grey, N+G6+G4, N+G6+G4- or N-G6-G4+), Epi-like cells (magenta, N+G6-G4-) and PrE-like cells (green-blue, N-G6+G4-, N-G6-G4+ or N-G6+G4+) of cells treated with CHX for the indicated time. **(B)** Average levels and standard error of NANOG (magenta), GATA6 (green), and GATA4 (blue) normalised to levels at 0h. Wild-type cells levels are in darker colours, and mutant cells levels are in lighter colours. Lines correspond to the best fit exponential decay model using the experimental data from wild-type(continuous) and mutant cells (dashed).
(TIF)

**S1 Table. Table with detailed information on the statistical tests.**
(XLSX)

**S1 Raw images. Raw images of gels and blots shown in S4 Fig.**
(PDF)

## Acknowledgments

Lesley Moore in the Animal Facility at the University of Bath, Christian Schröter for the GATA6 inducible cells. Alfonso Martinez Arias and Kat Hadjantonakis during the very early stage of this work. Cristina Merino for help in the schemes shown in Fig S4B, C and F. Michael Eibl for annotating ICM chimeric organoids. Imaging at the IUIBS (Instituto de Investigaciones Biomédicas y Sanitarias) was done at the SIMACE (Servicio de Investigación en Microscopía Avanzada Confocal y Electrónica). Mouse work at the IUIBS was done at the now established SABiA (Servicio de Animalario y Bienestar Animal).

## Author contributions

**Conceptualization:** Silvia Muñoz-Descalzo.

**Data curation:** Sabine C. Fischer, Silvia Muñoz-Descalzo.

**Formal analysis:** Joaquin Lilao-Garzon, Sabine C. Fischer, Jose Guillen, Silvia Muñoz-Descalzo.

**Funding acquisition:** Silvia Muñoz-Descalzo.

**Investigation:** Joaquin Lilao-Garzon, Elena Corujo-Simon, Meritxell Vinyoles, Sabine C. Fischer, Jose Guillen, Tina Balayo, Silvia Muñoz-Descalzo.

**Methodology:** Silvia Muñoz-Descalzo.

**Project administration:** Silvia Muñoz-Descalzo.

**Resources:** Silvia Muñoz-Descalzo.

**Software:** Sabine C. Fischer, Jose Guillen.

**Supervision:** Silvia Muñoz-Descalzo.

**Validation:** Silvia Muñoz-Descalzo.

**Visualization:** Silvia Muñoz-Descalzo.

**Writing – original draft:** Joaquin Lilao-Garzon, Silvia Muñoz-Descalzo.

**Writing – review & editing:** Joaquin Lilao-Garzon, Elena Corujo-Simon, Meritxell Vinyoles, Sabine C. Fischer, Jose Guillen, Silvia Muñoz-Descalzo.

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
