## [Decision Letter · Decision Letter 0]

20 Jul 2025

PONE-D-25-32872Wnt/β-catenin signalling assists cell fate decision making in the early mouse embryoPLOS ONE

Dear Dr. Muñoz-Descalzo,

Thank you for submitting your manuscript to PLOS ONE. After careful consideration, we feel that it has merit but does not fully meet PLOS ONE’s publication criteria as it currently stands. Therefore, we invite you to submit a revised version of the manuscript that addresses the points raised during the review process.

We look forward to receiving your revised manuscript.

Kind regards,

Gregory M. Kelly, PhD

Academic Editor

PLOS ONE

Journal Requirements:

3. To comply with PLOS ONE submissions requirements, in your Methods section, please provide additional information regarding the experiments involving animals and ensure you have included details on (1) methods of sacrifice, and (2) efforts to alleviate suffering

4. In the online submission form, you indicated that the data underlying the results presented in the study are available from the corresponding author upon request.

6. Please expand the acronym “ULPGC, ACIISI” (as indicated in your financial disclosure) so that it states the name of your funders in full.

7. Thank you for stating the following in the Acknowledgments Section of your manuscript:

Lesley Moore in the Animal Facility at the University of Bath, Christian Schröter for the GATA6 inducible cells. Alfonso Martinez Arias and Kat Hadjantonakis during the very early stage of this work. Cristina Merino for help in the schemes shown in Fig S4B, C and F. Michael Eibl for annotating ICM chimeric organoids. Early work at the University of Bath was supported by a Wellcome Trust Seed Award (109589/Z/15/Z) and ECS funding was provided by the University of Bath. At the ULPGC, work at SMDlab was funded by the ACIISI (CEI2019-02), Programa de Ayudas a la Investigación de la ULPGC, and ACIISI co-funded by FEDER Funds (ProID2020010013). JLG was supported by the ULPGC predoctoral program and SMD was supported by the ‘‘Viera y Clavijo’’ Program from the ACIISI, and the ULPGC. Imaging at the IUIBS was done at the SIMACE (Servicio de Investigación en Microscopía Avanzada Confocal y Electrónica). Mouse work at the IUIBS was done at the now established SABiA (Servicio de Animalario y Bienestar Animal). TB was funded by a “Ayuda para Personal Técnico de Apoyo” from the Ministerio de Economía Industria y Competitividad (PTA2017-14230-I). Animal work at the ULPGC was done at the SABiA (Servicio Científico Servicio de Animalario y Bienestar Animal). Work at the SCF lab was supported through funding by the Deutsche Forschungsgemeinschaft (DFG, German Research Foundation) project number 470129398 and start-up funding by the University of Wuerzburg. MV funding was provided by EMBO Postdoctoral fellowships (ALTF 509-2015)

Early work at the University of Bath was supported by a Wellcome Trust Seed Award (109589/Z/15/Z) and ECS funding was provided by the University of Bath. At the ULPGC, work at SMDlab was funded by the ACIISI (CEI2019-02), Programa de Ayudas a la Investigación de la ULPGC, and ACIISI co-funded by FEDER Funds (ProID2020010013). JLG was supported by the ULPGC predoctoral program and SMD was supported by the ‘‘Viera y Clavijo’’ Program from the ACIISI, and the ULPGC. Imaging at the IUIBS was done at the SIMACE (Servicio de Investigación en Microscopía Avanzada Confocal y Electrónica). Mouse work at the IUIBS was done at the now established SABiA (Servicio de Animalario y Bienestar Animal). TB was funded by a “Ayuda para Personal Técnico de Apoyo” from the Ministerio de Economía Industria y Competitividad (PTA2017-14230-I). Animal work at the ULPGC was done at the SABiA (Servicio Científico Servicio de Animalario y Bienestar Animal). Work at the SCF lab was supported through funding by the Deutsche Forschungsgemeinschaft (DFG, German Research Foundation) project number 470129398 and start-up funding by the University of Wuerzburg. MV funding was provided by EMBO Postdoctoral fellowships (ALTF 509-2015).

8. Please amend your list of authors on the manuscript to ensure that each author is linked to an affiliation. Authors’ affiliations should reflect the institution where the work was done (if authors moved subsequently, you can also list the new affiliation stating “current affiliation:….” as necessary).

Additional Editor Comments :

I have deemed this MS publishable, but taking the comments from the two Reviewers and incorporating them for clarity and to strengthen the MS. It is generally a well-written MS, despite some errors in spelling and grammar, that was noted by both Reviewers. I feel that the "science" was sound and the study addresses a question re. the ICM, specifically the Primitive Endoderm involving Wnt-ß-catenin. You have used many models to highlight the need for Wnts working with FGF/ERK to promote the formation of PrE and I, and the readers, am/will be convinced by the evidence provided.

Reviewers' comments:

Reviewer's Responses to Questions

**Comments to the Author**

1. Is the manuscript technically sound, and do the data support the conclusions?

Reviewer #1: Yes

Reviewer #2: Partly

2. Has the statistical analysis been performed appropriately and rigorously? 

Reviewer #1: Yes

Reviewer #2: Yes

3. Have the authors made all data underlying the findings in their manuscript fully available?

Reviewer #1: Yes

Reviewer #2: Yes

4. Is the manuscript presented in an intelligible fashion and written in standard English?

Reviewer #1: Yes

Reviewer #2: Yes

5. Review Comments to the Author

Reviewer #1: During mouse preimplantation development, the inner cell mass (ICM) differentiates into epiblast (Epi) cells, which express NANOG, and primitive endoderm (PrE) cells, which express GATA6, SOX17, and GATA4. The mechanisms controlling this cell fate decision are not fully understood. While FGF/ERK signalling is known to play a major role, it does not explain the coexpression of NANOG and GATA6, nor does it clarify how the fate choice is initiated. This study (PONE-D-25-32872) by Dr. Muñoz-Descalzo’s group investigates whether Wnt/β-catenin signalling also affects this process. Using two in vitro models—2D (flat-cultured cells) and 3D (ICM organoids)—along with mouse preimplantation embryo experiments, chemical and genetic approaches, and quantitative 3D immunofluorescence, the authors demonstrate a dual role for Wnt/β-catenin signalling. Specifically, β-catenin works in conjunction with FGF/ERK to promote PrE differentiation and stabilizes GATA6 and GATA4, thereby further supporting this cell fate decision. In summary, activation of the Wnt/β-catenin pathway promotes PrE formation, whereas its inhibition hinders it.

This is an elegant study that uses multiple cell and embryo models, along with chemical and genetic methods, to show β-catenin’s role in ICM cell fate. Only minor revisions are suggested to enhance the paper’s clarity.

Abstract, line 32; GATA6

Abstract, lines 39-41. It is recommended to combine the last two sentences of the abstract. Pathway activation is without context. Wnt/β-catenin activation promotes PrE differentiation.

Introduction, lines 47-49. Poor grammar, please revise.

Results, lines 105-106 and Figure 2. Do you mean that membrane b-catenin levels are higher among adjacent N+G6- cells than between adjacent N+G6- and N-G6+ or N-G6+ cells? Please revise the statement. The colours on the different cell types are missing in 2C.

Line 140; …in the presence of Wnt or XAV… do you mean Chi instead of Wnt?

Lines 156-160. How can N-G4+ cells be both PrE-like cells and Epi-like cells? Please clarify

Lines 195-199. I agree with you that the disparity in some of your results is likely attributed to differences in the developmental stages represented by the two in vitro systems. Could you elaborate further in the discussion, specifically discussing naïve and primed states and their impact on cell fate decisions? Did you ever treat your cells with the 2i (GSK3b, MEK inhibitor) cocktail or thought of using EpiSCs prior to other manipulations/evaluations?

Line 274,…measured expression levels….do you mean protein levels? This is a more accurate statement. You did not measure expression levels.

Discussion, line 283;…given the embryonic origin of these [cells] [25-27].

Reviewer #2: Overall Recommendation

Minor Revision. The study addresses an important and long-standing question, whether Wnt/β-catenin contributes to the ICM → Epiblast vs Primitive Endoderm (PrE) decision in the mouse blastocyst. The authors combine 2D inducible Gata6 mESC assays, 3D ICM organoids, and in vivo embryo manipulations. Quantitative imaging is a strength. While the study addresses an important question, the evidence supporting the central mechanistic claims (that β-catenin promotes PrE, cooperates with FGF/MAPK, and stabilizes GATA6/GATA4) would benefit from additional strengthening. In particular, clearer statistical analysis, transparent definition of biological replication, full data availability, and key experimental controls (signal specificity, temporal resolution, assessment of possible off-target small-molecule effects, and biochemical validation of the proposed stability mechanism) are needed to fully substantiate the conclusions.

Brief Summary of the Work

The authors quantify β-catenin localization across blastocyst stages and report higher membrane β-catenin in Epi precursors relative to PrE. They modulate Wnt/β-catenin chemically (CHIR, Wnt3a, XAV) and genetically (β-catenin CRISPR in tet::Gata6-mCherry mESCs; materno-zygotic β-catenin mutant embryos). Across 2D, 3D organoid, and embryo contexts, pathway activation is reported to increase PrE markers (SOX17, GATA4), whereas inhibition reduces PrE and/or delays differentiation. Rescue experiments with exogenous FGF suggest cooperation between Wnt/β-catenin and FGF/MAPK. Cycloheximide chase assays are used to infer that β-catenin stabilizes GATA6/GATA4 proteins, providing a post-transcriptional mechanism for PrE promotion.

Major Strengths

1. Multi-system approach (cells → organoids → embryos): Using complementary models is valuable for teasing apart context-dependent effects in early development.

2. Quantitative imaging & population analyses: Consistent use of quantitative immunofluorescence (QIF), automated segmentation (MINS referenced), and population stratification by marker combinations is a methodological strength vs prior qualitative studies.

3. Functional perturbations (chemical + genetic): Parallel small-molecule and CRISPR loss-of-function strategies increase interpretability if properly controlled.

4. Attempt to integrate with FGF/MAPK signalling: Testing for pathway interaction is a logical and timely extension given the dominance of FGF/ERK in PrE literature.

Major Concerns

1. Unit of Biological Replication & Statistical Inference

Across figures, cells are pooled for statistical tests (e.g., Z-tests on percentages, Mann-Whitney on single-cell intensities) without clearly accounting for embryo/organoid/plate as the experimental unit. This risks pseudo-replication and inflated significance. Please clarify, re-analyze using per-embryo/per-organoid summary values (biological n), and provide the number of independent litters/derivations.

2. Statistical Methods & Reporting

The manuscript mixes Z-tests, Mann-Whitney tests, and ad hoc p-threshold notations (e.g., *p<0.1; **p<0.05 in Fig.3 legend). The rationale for one- vs two-tailed, and multiple-comparison corrections (Bonferroni sometimes, not others) is inconsistent. Provide a unified statistical analysis section detailing: test choice; assumptions; whether data met assumptions; how multiple comparisons were handled; and exact p-values. Consider generalized linear mixed models for categorical lineage outcomes (PrE/Epi/ICM) with treatment as fixed and embryo as random effect.

3. Drug Concentrations, Specificity, and Timing

CHIR99021 (3 µM), Wnt3a (200 ng/ml), XAV939 (1 µM), FGFRi AZD4547 (1 µM), PD035901 (1 µM) are used; however, the manuscript lacks justification for doses relative to published EC50/IC50 in embryos, potential toxicity, and off-target effects especially given the small embryo volume and 24h exposures. Were dose–response curves performed?

Minor Points by Section

Title / Abstract

• Good scope; consider tempering “assists cell fate decision making” to “modulates efficiency/timing of PrE differentiation,” as data show delays and context-dependence rather than an absolute requirement.

Introduction

• Nicely situates FGF/MAPK primacy; however, the logic leap from mESC β-catenin roles to in vivo PrE promotion warrants clearer mechanistic hypotheses up front (adhesion vs transcription).

• Correct “GAT6” typo in Abstract; should be GATA6.

Results: Embryo Treatments

• Report how many embryos per litter; randomization and blinding of treatment; whether litter effects were controlled.

• Provide dpERK staining to confirm FGF pathway engagement across genotypes; otherwise interpretation that FGF bypasses β-catenin remains speculative.

• Methods say CHX 40 µM; please justify concentration and show that global translation inhibition is comparable across genotypes (e.g., puromycin incorporation assay).

Discussion

• Discussion currently weaves speculative oscillatory MAPK models (based on external refs) with current data; separate what is shown here from what is hypothesized.

• Reconcile contradictory scRNAseq finding that canonical Wnt targets are enriched in Epi cells with your conclusion that β-catenin promotes PrE; perhaps emphasize non-canonical / structural roles.

Methods: Imaging & Analysis

• State microscope settings (laser power, gain) ranges and whether identical across treatment groups; intensity comparisons otherwise problematic.

Methods: Embryo Culture

• Provide medium change schedule, embryo density per drop, and whether maternal genotype or litter order influenced developmental rate.

Compliance Statements

• Ethics statement is present; ensure it also appears in the Methods, per PLOS guidelines.

Presentation

• Standardize marker nomenclature (e.g., “N-G6+” vs “N−G6+”; minus signs).

• Ensure consistent superscripts/subscripts in genotype notation (β-catenin^+/−).

• Replace “stalls it” in Abstract conclusion with a more precise description (“reduces efficiency and delays progression”).

• Remove threshold p<0.1 significance coding; reserve symbols for conventional cutoffs or report exact p.

6. PLOS authors have the option to publish the peer review history of their article (what does this mean? ). If published, this will include your full peer review and any attached files.

**Do you want your identity to be public for this peer review?** For information about this choice, including consent withdrawal, please see our Privacy Policy .

Reviewer #1: No

Reviewer #2: No

---

## [Author Response · Author response to Decision Letter 1]

15 Dec 2025

The responses to the reviewer´s comments can be found at the "Response to reviewers" file

---

## [Decision Letter · Decision Letter 1]

19 Feb 2026

Wnt/β-catenin signalling modulates the timing of cell fate decision making in the early mouse embryo

PONE-D-25-32872R1

Dear Dr. Muñoz-Descalzo,

many, many apologies, I feel we have straightened things out.  We’re pleased to inform you that your manuscript has been judged scientifically suitable for publication and will be formally accepted for publication once it meets all outstanding technical requirements.

Kind regards,

Gregory M. Kelly, PhD

Academic Editor

PLOS One

Additional Editor Comments (optional):

Reviewers' comments:

Reviewer's Responses to Questions

**Comments to the Author**

1. If the authors have adequately addressed your comments raised in a previous round of review and you feel that this manuscript is now acceptable for publication, you may indicate that here to bypass the “Comments to the Author” section, enter your conflict of interest statement in the “Confidential to Editor” section, and submit your "Accept" recommendation.

Reviewer #2: All comments have been addressed

2. Is the manuscript technically sound, and do the data support the conclusions?

Reviewer #2: Yes

3. Has the statistical analysis been performed appropriately and rigorously? 

Reviewer #2: Yes

4. Have the authors made all data underlying the findings in their manuscript fully available?

Reviewer #2: Yes

5. Is the manuscript presented in an intelligible fashion and written in standard English?

Reviewer #2: Yes

6. Review Comments to the Author

Reviewer #2: The authors have appropriately addressed the previously raised concerns. The Materials and Methods section now clearly defines experimental units, includes measures to avoid litter effects, and clarifies inclusion criteria. Statistical analyses are presented in a unified and transparent manner, with consistent justification of test selection, tail usage, and multiple-comparison corrections. Drug concentrations and treatment timing are now clearly justified with reference to published studies. These revisions significantly improve the rigor and interpretability of the work.

7. PLOS authors have the option to publish the peer review history of their article (what does this mean? ). If published, this will include your full peer review and any attached files.

**Do you want your identity to be public for this peer review?** For information about this choice, including consent withdrawal, please see our Privacy Policy .

Reviewer #2: **Yes:** Nuwanthika Wathuliyadde

---

## [Editor Report · Acceptance letter]

PONE-D-25-32872R1

PLOS One

Dear Dr. Muñoz-Descalzo,

I'm pleased to inform you that your manuscript has been deemed suitable for publication in PLOS One. Congratulations! Your manuscript is now being handed over to our production team.

Kind regards,

on behalf of

Dr Gregory M. Kelly

Academic Editor

PLOS One